# Ribosomal RNA (rRNA) sequences from 33 globally distributed mosquito species for improved metagenomics and species identification

Cassandra Koh[1]*, Lionel Frangeul[1], Hervé Blanc[1], Carine Ngoagouni[2], Sébastien Boyer[3], Philippe Dussart[4], Nina Grau[5], Romain Girod[5], Jean-Bernard Duchemin[6], Maria-Carla Saleh[1]*

[1]Institut Pasteur, Université Paris Cité, CNRS UMR3569, Viruses and RNA Interference Unit, F-75015, Paris, France; [2]Institut Pasteur de Bangui, Medical Entomology Laboratory, Bangui, Central African Republic; [3]Institut Pasteur du Cambodge, Medical and Veterinary Entomology Unit, Phnom Penh, Cambodia; [4]Institut Pasteur du Cambodge, Virology Unit, Phnom Penh, Cambodia; [5]Institut Pasteur de Madagascar, Medical Entomology Unit, Antananarivo, Madagascar; [6]Institut Pasteur de la Guyane, Vectopôle Amazonien Emile Abonnenc, Cayenne, French Guiana

*For correspondence:
cassandra.koh@pasteur.fr (CK);
carla.saleh@pasteur.fr (MCS)

Competing interest: The authors declare that no competing interests exist.

**Abstract** Total RNA sequencing (RNA-seq) is an important tool in the study of mosquitoes and the RNA viruses they vector as it allows assessment of both host and viral RNA in specimens. However, there are two main constraints. First, as with many other species, abundant mosquito ribosomal RNA (rRNA) serves as the predominant template from which sequences are generated, meaning that the desired host and viral templates are sequenced far less. Second, mosquito specimens captured in the field must be correctly identified, in some cases to the sub-species level. Here, we generate mosquito rRNA datasets which will substantially mitigate both of these problems. We describe a strategy to assemble novel rRNA sequences from mosquito specimens and produce an unprecedented dataset of 234 full-length 28S and 18S rRNA sequences of 33 medically important species from countries with known histories of mosquito-borne virus circulation (Cambodia, the Central African Republic, Madagascar, and French Guiana). These sequences will allow both physical and computational removal of rRNA from specimens during RNA-seq protocols. We also assess the utility of rRNA sequences for molecular taxonomy and compare phylogenies constructed using rRNA sequences versus those created using the gold standard for molecular species identification of specimens—the mitochondrial *cytochrome* c *oxidase I* (COI) gene. We find that rRNA- and COI-derived phylogenetic trees are incongruent and that 28S and concatenated 28S+18S rRNA phylogenies reflect evolutionary relationships that are more aligned with contemporary mosquito systematics. This significant expansion to the current rRNA reference library for mosquitoes will improve mosquito RNA-seq metagenomics by permitting the optimization of species-specific rRNA depletion protocols for a broader range of species and streamlining species identification by rRNA sequence and phylogenetics.

## Editor's evaluation

Mosquitoes are an important vector for viruses and other pathogens worldwide. However, significant genomic resources are scarce for the study of these species. In this work, the authors create a significant genomic resource that will enable the study of mosquitoes and the pathogens that they carry.

## Introduction

Mosquitoes top the list of vectors for arthropod-borne diseases, being implicated in the transmission of many human pathogens responsible for arboviral diseases, malaria, and lymphatic filariasis (*WHO, 2017*). Mosquito-borne viruses circulate in sylvatic (between wild animals) or urban (between humans) transmission cycles driven by different mosquito species with their own distinct host preferences. Although urban mosquito species are chiefly responsible for amplifying epidemics in dense human populations, sylvatic mosquitoes maintain the transmission of these viruses among forest-dwelling animal reservoir hosts and are involved in spillover events when humans enter their ecological niches (*Valentine et al., 2019*). Given that mosquito-borne virus emergence is preceded by such spillover events, continuous surveillance and virus discovery in sylvatic mosquitoes is integral to designing effective public health measures to pre-empt or respond to mosquito-borne viral epidemics.

Metagenomics on field specimens is a powerful method in our toolkit to understand mosquito-borne disease ecology through the One Health lens (*Webster et al., 2016*). With next-generation sequencing becoming more accessible, such studies have provided unprecedented insights into the interfaces among mosquitoes, their environment, and their animal and human hosts. As mosquito-associated viruses are mostly RNA viruses, RNA sequencing (RNA-seq) is especially informative for surveillance and virus discovery. However, working with lesser studied mosquito species poses several problems.

First, metagenomics studies based on RNA-seq are bedevilled by overabundant ribosomal RNAs (rRNAs). These non-coding RNA molecules comprise at least 80% of the total cellular RNA population (*Gale and Crampton, 1989*). Due to their length and their abundance, they are a sink for precious next-generation sequencing reads, decreasing the sensitivity of pathogen detection unless depleted during library preparation. Yet the most common rRNA depletion protocols require prior knowledge of rRNA sequences of the species of interest as they involve hybridizing antisense oligos to the rRNA molecules prior to removal by ribonucleases (*Fauver et al., 2019*; *Phelps et al., 2021*) or by bead capture (*Kukutla et al., 2013*). Presently, reference sequences for rRNAs are limited to only a handful of species from three genera: *Aedes*, *Culex*, and *Anopheles* (*Ruzzante et al., 2019*). The lack of reliable rRNA depletion methods could deter mosquito metagenomics studies from expanding their sampling diversity, resulting in a gap in our knowledge of mosquito vector ecology. The inclusion of lesser studied yet medically relevant sylvatic species is therefore imperative.

Second, species identification based on morphology is notoriously complicated for members of certain species subgroups. This is especially the case among *Culex* subgroups. Sister species are often sympatric and show at least some competence for a number of viruses, such as Japanese encephalitis virus, St Louis encephalitic virus, and Usutu virus (*Nchoutpouen et al., 2019*). Although they share many morphological traits, each of these species have distinct ecologies and host preferences, thus the challenge of correctly identifying vector species can affect epidemiological risk estimation for these diseases (*Farajollahi et al., 2011*). DNA molecular markers are often employed to a limited degree of success to distinguish between sister species (*Batovska et al., 2017*; *Zittra et al., 2016*).

To address the lack of full-length rRNA sequences in public databases, we sought to determine the 28S and 18S rRNA sequences of a diverse set of Old and New World sylvatic mosquito species from four countries representing three continents: Cambodia, the Central African Republic, Madagascar, and French Guiana. These countries, due to their proximity to the equator, contain high mosquito biodiversity (*Foley et al., 2007*) and have had long histories of mosquito-borne virus circulation (*Desdouits et al., 2015*; *Halstead, 2019*; *Héraud et al., 2022*; *Jacobi and Serie, 1972*; *Ratsito-rahina et al., 2008*; *Saluzzo et al., 2017*; *Zeller et al., 2016*). Increased and continued surveillance of local mosquito species could lead to valuable insights on mosquito virus biogeography. Using a unique score-based read filtration strategy to remove interfering non-mosquito rRNA reads for accurate de novo assembly, we produced a dataset of 234 novel full-length 28S and 18S rRNA sequences from 33 mosquito species, 30 of which have never been recorded before.

We also explored the functionality of 28S and 18S rRNA sequences as molecular markers by comparing their performance to that of the mitochondrial *cytochrome c oxidase subunit I* (COI) gene for molecular taxonomic and phylogenetic investigations. The COI gene is the most widely used DNA marker for molecular species identification and forms the basis of the Barcode of Life Data System (BOLD) (*Hebert et al., 2003*; *Ratnasingham and Hebert, 2007*). Presently, full-length rRNA sequences are much less represented compared to other molecular markers. However, given the

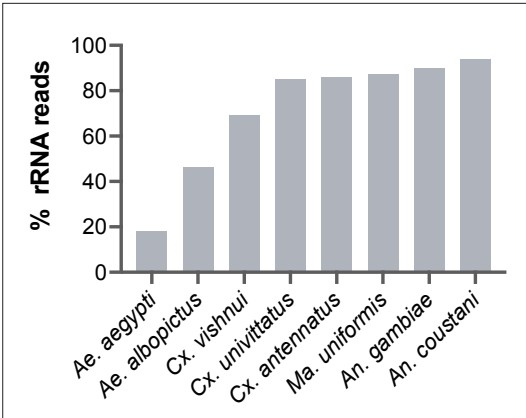

**Figure 1.** Percentage of rRNA reads in mosquito total RNA sequencing (RNA-seq) data after depletion using probes antisense to *Aedes aegypti* sequences. Pools of five individual mosquitoes from genera *Aedes* (*Ae*), *Culex* (*Cx*), *Mansonia* (*Ma*), and *Anopheles* (*An*) were ribodepleted by probe hybridisation followed by RNase H digestion according to the protocol by *Morlan et al., 2012*. Y-axis depicts percentages of remaining rRNA reads calculated as the number of rRNA reads over total reads per sample pool. Depletion efficiency decreases with taxonomic distance from *Ae. aegypti* underlining the need for reference sequences for species of interest.

availability of relevant reference sequences, 28S and concatenated 28S+18S rRNA sequences can be the better approach for molecular taxonomy and phylogenetic studies. We hope that our sequence dataset, with its species diversity and eco-geographical breadth, and the assembly strategy we describe would further facilitate the use of rRNA as markers. In addition, this dataset enables the design of species-specific oligos for cost-effective rRNA depletion for a broader range of mosquito species and streamlined molecular species identification during RNA-seq.

## Results

### Poor rRNA depletion using a non-specific depletion method

During library preparations of mosquito samples for RNA-seq, routinely used methods for depleting rRNA are commercial kits optimised for human or mice samples (*Belda et al., 2019*; *Bishop-Lilly et al., 2010*; *Chandler et al., 2015*; *Kumar et al., 2012*; *Weedall et al., 2015*; *Zakrzewski et al., 2018*) or through 80–100 base pair antisense probe hybridisation followed by ribonuclease digestion (*Fauver et al., 2019*; *Phelps et al., 2021*). In cases where the complete reference rRNA sequence of the target species is not known, oligos would be designed based on the rRNA sequence of the closest related species (25, this study). These methods should deplete reads from the conserved regions of rRNA sequences. However, reads from the variable regions remain at abundances high enough to compromise RNA-seq output. In our hands, we have found that using probes designed for the *Ae. aegypti* rRNA sequence followed by RNase H digestion according to the protocol published by *Morlan et al., 2012*, produced poor depletion in *Aedes albopictus*, and in Culicine and Anopheline species (*Figure 1*), in which between 46% and 94% of reads post-depletion were ribosomal. Additionally, the lack of full-length reference rRNA sequences compromises the *in silico* clean-up of remaining rRNA reads from sequencing data, as reads belonging to variable regions would not be removed. To solve this and to enable RNA-seq metagenomics on a broader range of mosquito species, we performed RNA-seq to generate reference rRNA sequences for 33 mosquito species representing 10 genera from Cambodia, the Central African Republic, Madagascar, and French Guiana. Most of these species are associated with vector activity for various pathogens in their respective ecologies (*Table 1*). In parallel, we sequenced the mitochondrial COI gene to perform molecular species identification of our samples and to comparatively evaluate the use of rRNA as a molecular marker (*Figure 2*).

### rRNA reads filtering and sequence assembly

Assembling Illumina reads to reconstruct rRNA sequences from total mosquito RNA is not a straightforward task. Apart from host rRNA, total RNA samples also contain rRNA from other organisms associated with the host (microbiota, external parasites, or ingested diet). As rRNA sequences share high homology in conserved regions, Illumina reads (150 bp) from non-host rRNA can interfere with the contig assembly of host 28S and 18S rRNA.

Our score-based filtration strategy, described in detail in the Materials and methods section, allowed us to bioinformatically remove interfering rRNA reads and achieve successful de novo assembly of 28S and 18S rRNA sequences for all our specimens. Briefly, for each Illumina read, we computed a ratio of BLAST scores against an Insecta library over scores against a Non-Insecta library (*Figure 2A*).

**Table 1.** Mosquito species represented in this study and their vector status.

| Mosquito taxonomy[‡] | Origin* | Collection site (ecosystem type) | Vector for[†] | Reference |
|---|---|---|---|---|
| *Aedes (Fredardsius) vittatus* | CF | Rural (village) | ZIKV, CHIKV, YFV | *Diallo et al., 2020* |
| *Aedes (Ochlerotatus) scapularis* | GF | Rural (village) | YFV | *Vasconcelos et al., 2001* |
| *Aedes (Ochlerotatus) serratus* | GF | Rural (village) | YFV, OROV | *Cardoso et al., 2010*; *Romero-Alvarez and Escobar, 2018* |
| *Aedes (Stegomyia) aegypti* | CF | Urban | DENV, ZIKV, CHIKV, YFV | *Kraemer et al., 2019* |
| *Aedes (Stegomyia) albopictus* | CF, KH | Rural (village, nature reserve) | DENV, ZIKV, CHIKV, YFV, JEV | *Auerswald et al., 2021*; *Kraemer et al., 2019* |
| *Aedes (Stegomyia) simpsoni* | CF | Rural (village) | YFV | *Mukwaya et al., 2000* |
| *Anopheles (Anopheles) baezai* | KH | Rural (nature reserve) | Unreported | – |
| *Anopheles (Anopheles) coustani* | MG, CF | Rural (village) | RVFV, malaria | *Mwangangi et al., 2013*; *Nepomichene et al., 2018*; *Ratovonjato et al., 2011* |
| *Anopheles (Cellia) funestus* | MG, CF | Rural (village) | ONNV, malaria | *Lutomiah et al., 2013*; *Tabue et al., 2017* |
| *Anopheles (Cellia) gambiae* | MG, CF | Rural (village) | ONNV, malaria | *Brault et al., 2004* |
| *Anopheles (Cellia) squamosus* | MG | Rural (village) | RVFV, malaria | *Ratovonjato et al., 2011*; *Stevenson et al., 2016* |
| *Coquillettidia (Rhynchotaenia) venezuelensis* | GF | Rural (village) | OROV | *Travassos da Rosa et al., 2017* |
| *Culex (Culex) antennatus* | MG | Rural (village) | RVFV | *Nepomichene et al., 2018*; *Ratovonjato et al., 2011* |
| *Culex (Culex) duttoni* | CF | Rural (village) | Unreported | – |
| *Culex (Culex) neavei* | MG | Rural (village) | USUV | *Nikolay et al., 2011* |
| *Culex (Culex) orientalis* | KH | Rural (nature reserve) | JEV | *Kim et al., 2015* |
| *Culex (Culex) perexiguus* | MG | Rural (village) | WNV, USUV | *Vezenegho et al., 2022* |
| *Culex (Culex) pseudovishnui* | KH | Rural (nature reserve) | JEV | *Auerswald et al., 2021* |
| *Culex (Culex) quinquefasciatus* | MG, CF, KH | Rural (village, nature reserve) | ZIKV, JEV, WNV, DENV, SLEV, RVFV, *Wuchereria bancrofti* | *Bhattacharya and Basu, 2016*; *Maquart et al., 2021*; *Ndiaye et al., 2016*; *Serra et al., 2016* |
| *Culex (Culex) tritaeniorhynchus* | MG, KH | Rural (village, nature reserve) | JEV, WNV, RVFV | *Auerswald et al., 2021*; *Hayes et al., 1980*; *Jupp et al., 2002* |
| *Culex (Melanoconion) spissipes* | GF | Rural (village) | VEEV | *Weaver et al., 2004* |
| *Culex (Melanoconion) portesi* | GF | Rural (village) | VEEV, TONV | *Talaga et al., 2021*; *Weaver et al., 2004* |
| *Culex (Melanoconion) pedroi* | GF | Rural (village) | EEEV, VEEV, MADV | *Talaga et al., 2021*; *Turell et al., 2008* |
| *Culex (Oculeomyia) bitaeniorhynchus* | MG, KH | Rural (village, nature reserve) | JEV | *Auerswald et al., 2021* |
| *Culex (Oculeomyia) poicilipes* | MG | Rural (village) | RVFV | *Ndiaye et al., 2016* |
| *Eretmapodites intermedius* | CF | Rural (village) | Unreported | – |
| *Limatus durhamii* | GF | Rural (village) | ZIKV | *Barrio-Nuevo et al., 2020* |
| *Mansonia (Mansonia) titillans* | GF | Rural (village) | VEEV, SLEV | *Hoyos-López et al., 2015*; *Turell, 1999* |
| *Mansonia (Mansonioides) indiana* | KH | Rural (nature reserve) | JEV | *Arunachalam et al., 2004* |
| *Mansonia (Mansonioides) uniformis* | MG, CF, KH | Rural (village, nature reserve) | RVFV, *Wuchereria bancrofti* | *Lutomiah et al., 2013*; *Ughasi et al., 2012* |
| *Mimomyia (Etorleptiomyia) mediolineata* | MG | Rural (village) | Unreported | – |

*Table 1 continued on next page*

*Table 1 continued*

| Mosquito taxonomy[‡] | Origin* | Collection site (ecosystem type) | Vector for[†] | Reference |
|---|---|---|---|---|
| *Psorophora (Janthinosoma) ferox* | GF | Rural (village) | ROCV | *Mitchell et al., 1986* |
| *Uranotaenia (Uranotaenia) geometrica* | GF | Rural (village) | Unreported | – |

*Dengue virus, DENV; Zika virus, ZIKV; chikungunya virus, CHIKV; Yellow Fever virus, YFV; Oropouche virus, OROV; Japanese encephalitis virus, JEV; Rift Valley Fever virus, RVFV; O'Nyong Nyong virus, ONNV; Usutu virus, USUV; West Nile virus, WNV; St Louis encephalitis virus, SLEV; Venezuelan equine encephalitis virus, VEEV; Tonate virus, TONV; Eastern equine encephalitis virus, EEEV; Madariaga virus, MADV; Rocio virus, ROCV.

[†]Origin countries are listed as their ISO alpha-2 codes: Central African Republic, CF; Cambodia, KH; Madagascar, MG; French Guiana, GF.

[‡]Subgenus indicated in brackets.

Based on their ratio of scores, reads could be segregated into four categories (*Figure 2B*): (i) reads mapping only to the Insecta library, (ii) reads mapping better to the Insecta relative to Non-Insecta library, (iii) reads mapping better to the Non-Insecta relative to the Insecta library, and (iv) reads mapping only to the Non-Insecta library. By applying a conservative threshold at 0.8 to account for the non-exhaustiveness of the SILVA database, we removed reads that likely do not originate from mosquito rRNA. Notably, 15 of our specimens were engorged with vertebrate blood, a rich source of non-mosquito rRNA (*Appendix 1—table 1*). The successful assembly of complete 28S and 18S rRNA sequences for these specimens demonstrates that this strategy performs as expected even with high amounts of non-host rRNA reads. This is particularly important in studies on field-captured mosquitoes as females are often sampled already having imbibed a blood meal or captured using the human landing catch technique.

We encountered challenges for three specimens morphologically identified as *Mansonia africana* (Specimen ID S33–S35) (*Appendix 1—table 1*). COI amplification by PCR did not produce any product, hence COI sequencing could not be used to confirm species identity. In addition, the genome assembler SPAdes (*Bankevich et al., 2012*) was only able to assemble partial length rRNA contigs, despite the high number of reads with high scores against the Insecta library. Among other *Mansonia* specimens, these partial length contigs shared the highest similarity with contigs obtained from sample 'Ma uniformis CF S51'. We then performed a guided assembly using the 28S and 18S sequences of this specimen as references, which successfully produced full-length contigs. In two of these specimens (Specimen ID S34 and S35), our assembly initially produced two sets of 28S and 18S rRNA sequences, one of which was similar to mosquito rRNA with low coverage and another with 10-fold higher coverage and 95% nucleotide sequence similarity to a water mite of genus *Horreolanus* known to parasitize mosquitoes. Our success in obtaining rRNA sequences for mosquito and water mite shows that our strategy can be applied to metabarcoding studies where the input material comprises multiple insect species, provided that appropriate reference sequences of the target species or of a close relative are available.

Altogether, we were able to assemble 122 28S and 114 18S full-length rRNA sequences for 33 mosquito species representing 10 genera sampled from four countries across three continents. This dataset contains, to our knowledge, the first records for 30 mosquito species and for seven genera: *Coquillettidia*, *Mansonia*, *Limatus*, *Mimomyia*, *Uranotaenia*, *Psorophora,* and *Eretmapodites*. Individual GenBank accession numbers for these sequences and specimen information are listed in *Appendix 1—table 1*.

## Comparative phylogeny of novel rRNA sequences relative to existing records

To verify the assembly accuracy of our rRNA sequences, we constructed a comprehensive phylogenetic tree from the full-length 28S rRNA sequences generated from our study and included relevant rRNA sequences publicly available from GenBank (*Figure 3*). We applied a search criterion for GenBank sequences with at least 95% coverage of our sequence lengths (~4000 bp), aiming to represent as many species or genera as possible. Although we rarely found records for the same species included in our study, the resulting tree showed that our 28S sequences generally clustered according to their respective species and subgenera, supported by moderate to good bootstrap support at terminal

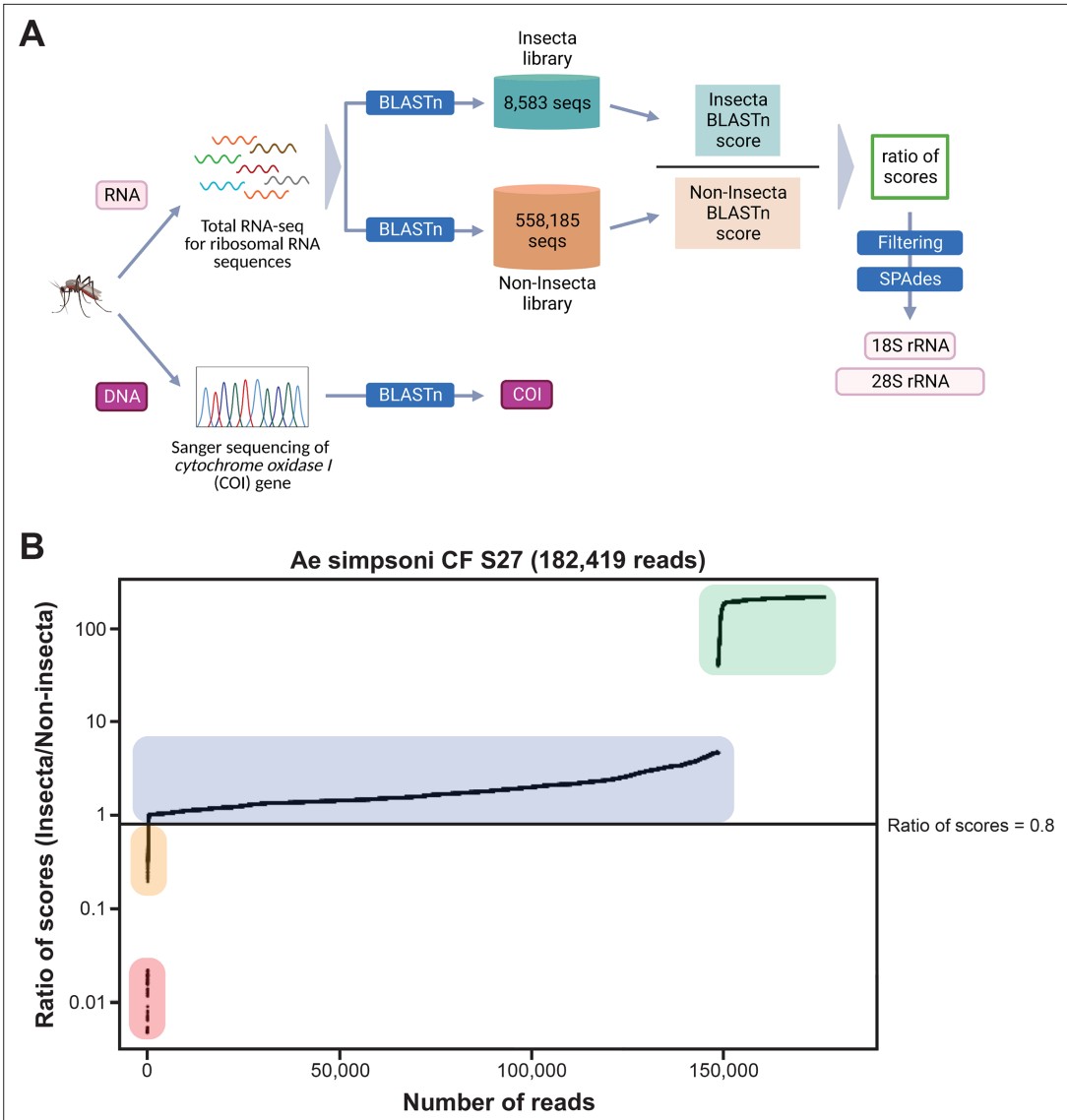

**Figure 2.** Novel mosquito rRNA sequences were obtained using a unique reads filtering method. (**A**) Schematic of sequencing and bioinformatics analyses performed in this study to obtain full-length 18S and 28S rRNA sequences as well as cytochrome *c* oxidase I (COI) DNA sequences. Nucleic acids were isolated from mosquito specimens for next-generation (for rRNA) or Sanger (for COI) sequencing. Two in-house libraries were created from the SILVA rRNA gene database: Insecta and Non-Insecta, which comprises 8,585 sequences and 558,185 sequences, respectively. Following BLASTn analyses against these two libraries, each RNA-sequencing (RNA-seq) read is assigned a ratio of BLASTn scores to describe their relative nucleotide similarity to insect rRNA sequences. Based on these ratios of scores, RNA-seq reads can then be filtered to remove non-mosquito reads prior to assembly with SPAdes to give full-length 18S and 28S rRNA sequences. Image created with https://biorender.com/. (**B**) Based on their ratio of scores, reads can be segregated into four categories, as shown on this ratio of scores versus number of reads plot for the representative specimen 'CF S27': (i) reads with hits only in the Insecta library (shaded in green), (ii) reads with a higher score against the Insecta library (shaded in blue), (iii) reads with a higher score against the Non-Insecta library (shaded in yellow), and (iv) reads with no hits in the Insecta library (shaded in red). We applied a conservative threshold at 0.8, indicated by the black horizontal line, where only reads above this threshold are used in the assembly with SPAdes. For this given specimen, 175,671 reads (96.3% of total reads) passed the ≥0.8 cut-off, 325 reads (0.18% of total reads) had ratios of scores <0.8, while 6,423 reads (3.52%) did not have hits against the Insecta library.

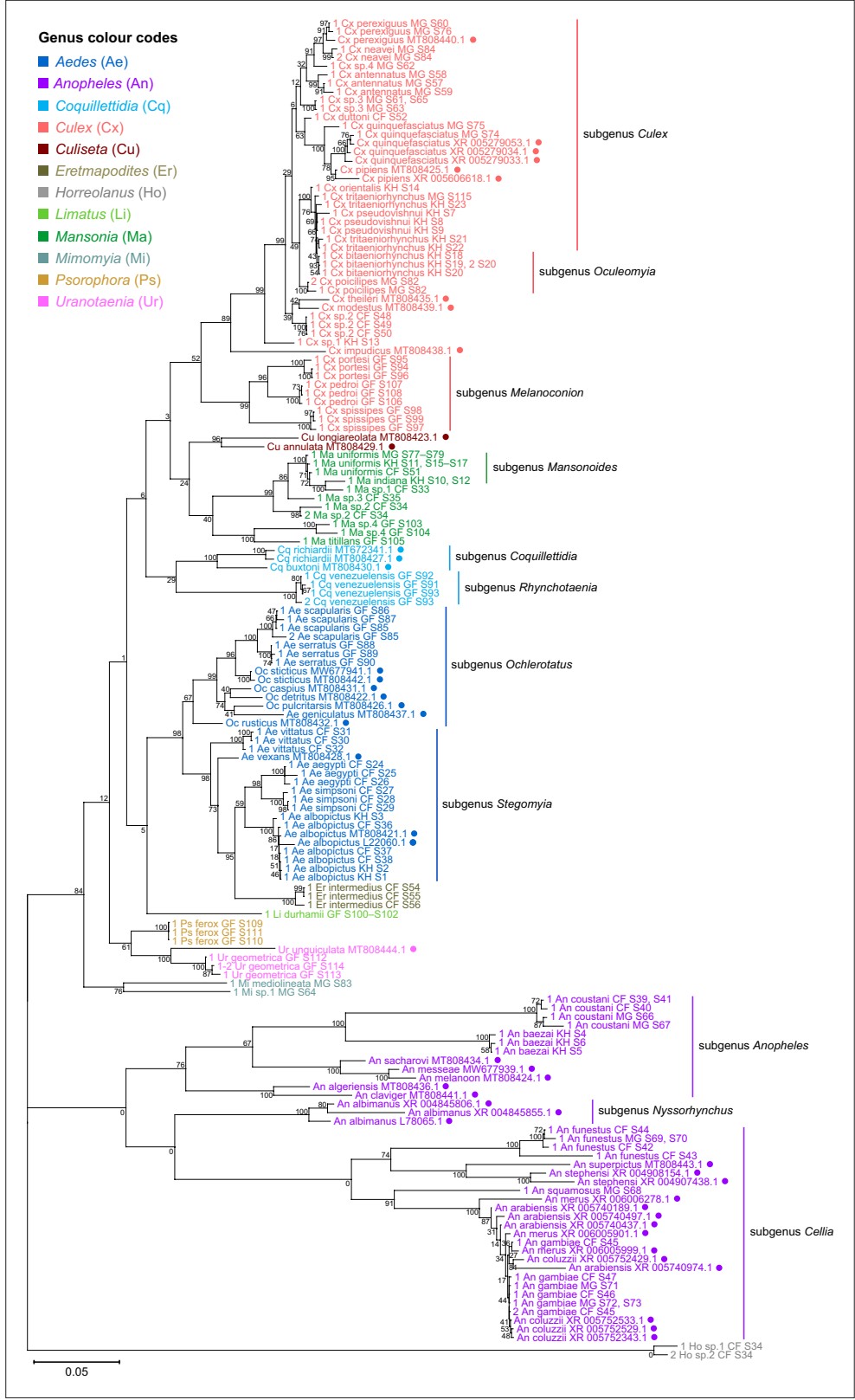

**Figure 3.** 28S sequences generated from this study clustered with conspecifics or congenerics from existing GenBank records. A rooted phylogenetic tree based on full-length 28S sequences (3,900 bp) from this study and from GenBank was inferred using the maximum-likelihood method and constructed to scale in MEGA X (**Kumar et al., 2018**) using an unknown *Horreolanus* species found among our samples as an outgroup. Values at each

*Figure 3 continued on next page*

*Figure 3 continued*

node indicate bootstrap support (%) from 500 replications. Sequences from GenBank are annotated with filled circles and their accession numbers are shown. For sequences from this study, each specimen label contains information on taxonomy, origin (in two-letter country codes), and specimen ID number. Some specimens produced up to two consensus 28S sequences; this is indicated by the numbers 1 or 2 at the beginning of the specimen label. Specimen genera are indicated by colour: *Culex* in coral, *Anopheles* in purple, *Aedes* in dark blue, *Mansonia* in dark green, *Culiseta* in maroon, *Limatus* in light green, *Coquillettidia* in light blue, *Psorophora* in yellow, *Mimomyia* in teal, *Uranotaenia* in pink, and *Eretmapodites* in brown. Scale bar at 0.05 is shown.

The online version of this article includes the following source data and figure supplement(s) for figure 3:

**Source data 1.** Multiple sequence alignment of 169 28S rRNA sequences from this study and from GenBank (FASTA).

**Figure supplement 1.** Interspecific and intersubgeneric distances within the genus *Anopheles* indicate a greater degree of divergence than those within any other genera of family Culicidae.

**Figure supplement 2.** Sequence conservation among 169 28S rRNA sequences obtained from this study and from GenBank combined.

nodes. Species taxa generally formed monophyletic clades, with the exception of *An. gambiae* and *Cx. quinquefasciatus. An. gambiae* 28S rRNA sequences formed a clade with closely related sequences from *Anopheles arabiensis, Anopheles merus*, and *Anopheles coluzzii,* suggesting unusually high interspecies homology for Anophelines or other members of subgenus *Cellia* (*Figure 3*, in purple, subgenus *Cellia*). Meanwhile, *Cx. quinquefasciatus* 28S rRNA sequences formed a taxon paraphyletic to sister species *Culex pipiens* (*Figure 3*, in coral, subgenus *Culex*).

28S rRNA sequence-based phylogenetic reconstructions (*Figure 3*, with GenBank sequences; *Figure 4—figure supplement 1*, this study only) showed marked incongruence to that of 18S rRNA sequences (*Figure 4—figure supplement 2*). Although all rRNA trees show the bifurcation of family Culicidae into subfamilies Anophelinae (genus *Anopheles*, in purple) and Culicinae (all other genera), the recovered intergeneric phylogenetic relationships vary between the 28S and 18S rRNA trees and are weakly supported. The 18S rRNA tree also exhibited several taxonomic anomalies: (i) the lack of definitive clustering by species within the *Culex* subgenus (in coral); (ii) the lack of distinction between 18S rRNA sequences of *Cx. pseudovishnui* and *Cx. tritaeniorhynchus* (in coral); (iii) the placement of Ma sp.3 CF S35 (in dark green) within a *Culex* clade; and (iv) the lack of a monophyletic *Mimomyia* clade (in teal) (*Figure 4—figure supplement 2*). However, 28S and 18S rRNA sequences are encoded by linked loci in rDNA clusters and should not be analysed separately.

Indeed, when concatenated 28S+18S rRNA sequences were generated from the same specimens (*Figure 4*), the phylogenetic tree resulting from these sequences more closely resembles the 28S tree (*Figure 3*) with regard to the basal position of the *Mimomyia* clade (in teal) within the Culicinae subfamily with good bootstrap support in either tree (84% in 28S rRNA tree, 100% in concatenated 28S+18S rRNA tree). For internal nodes, bootstrap support values were higher in the concatenated tree compared to the 28S tree. Interestingly, the 28S+18S rRNA tree formed an *Aedini* tribe-clade encompassing taxa from genera *Psorophora* (in yellow), *Aedes* (in dark blue), and *Eretmapodites* (in brown), possibly driven by the inclusion of 18S rRNA sequences. Concatenation also resolved the anomalies found in the 18S rRNA tree and added clarity to the close relationship between *Culex* (in coral) and *Mansonia* (in dark green) taxa. Of note, relative to the 28S tree (*Figure 3*) the *Culex* and *Mansonia* genera are no longer monophyletic in the concatenated 28S+18S rRNA tree (*Figure 4*). Genus *Culex* is paraphyletic with respect to subgenus *Mansonoides* of genus *Mansonia* (*Figure 3*). *Ma. titillans* and Ma sp.4, which we suspect to be *Mansonia pseudotitillans,* always formed a distinct branch in 28S or 18S rRNA phylogenies, thus possibly representing a clade of subgenus *Mansonia*.

The concatenated 28S+18S rRNA tree (*Figure 4*) recapitulates what is classically known about the systematics of our specimens, namely (i) the early divergence of subfamily Anophelinae from subfamily Culicinae, (ii) the division of genus *Anopheles* (in purple) into two subgenera, *Anopheles* and *Cellia*, (iii) the division of genus *Aedes* (in dark blue) into subgenera *Stegomyia* and *Ochlerotatus*, (iv) the divergence of the monophyletic subgenus *Melanoconion* within the *Culex* genus (in coral) (*Harbach, 2007*; *Harbach and Kitching, 2016*).

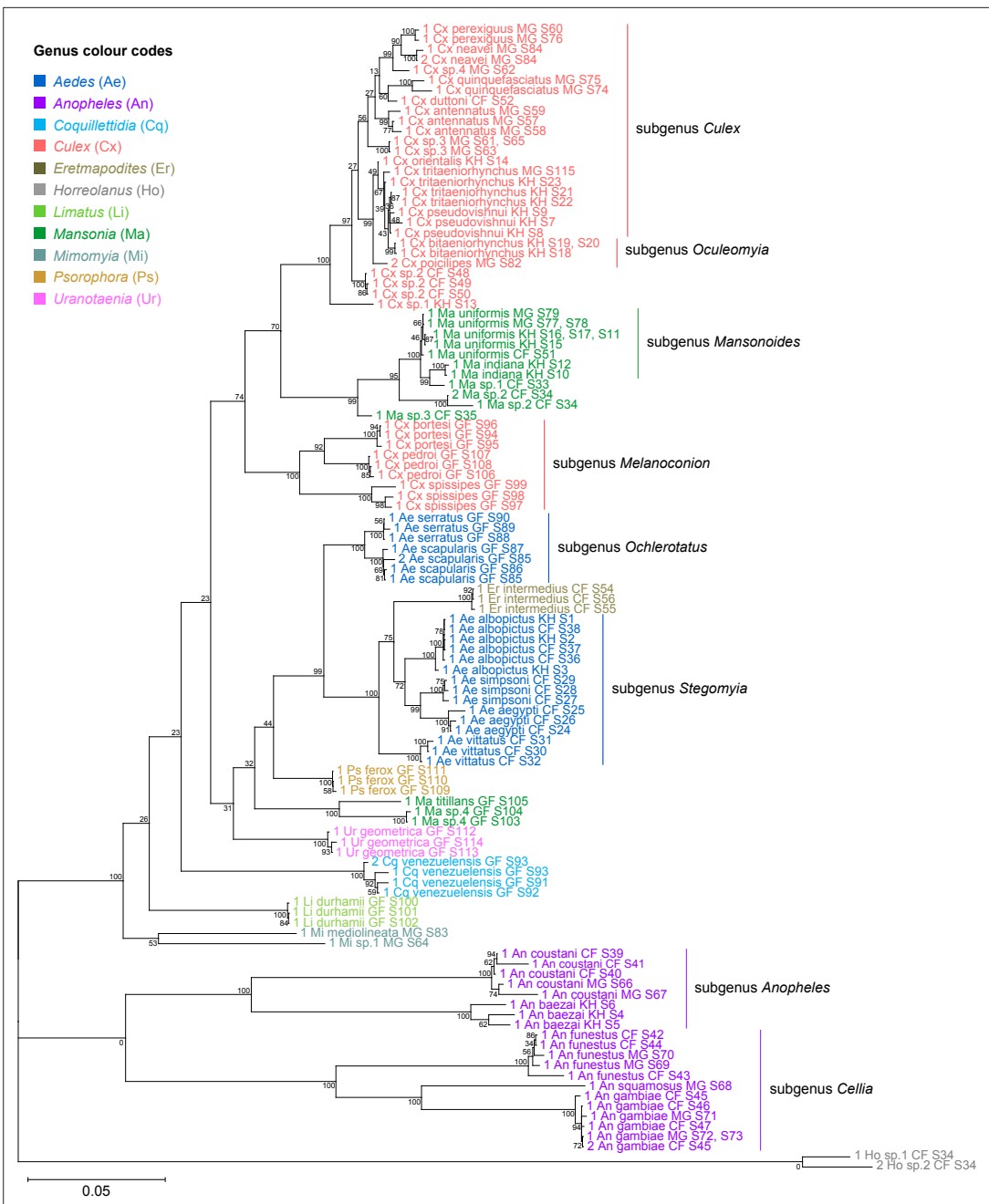

**Figure 4.** Concatenating 28S and 18S rRNA sequences produces phylogenetic relationships that are concordant with classical Culicidae systematics with higher bootstrap support than 28S sequences alone. This phylogenetic tree based on concatenated 28S+18S rRNA sequences (3,900+1,900 bp) generated from this study was inferred using the maximum-likelihood method and constructed to scale using MEGA X (*Kumar et al., 2018*) using an unknown *Horreolanus* species found among our samples as an outgroup. Values at each node indicate bootstrap support (%) from 500 replications. Each specimen label contains information on taxonomy, origin (as indicated in two-letter country codes), and specimen ID number. Some specimens produced up to two consensus 28S+18S rRNA sequences; this is indicated by the numbers 1 or 2 at the beginning of the specimen label. Specimen genera are indicated by colour: *Culex* in coral, *Anopheles* in purple, *Aedes* in dark blue, *Mansonia* in dark green, *Limatus* in light green, *Coquillettidia* in light blue, *Psorophora* in yellow, *Mimomyia* in teal, *Uranotaenia* in pink, and *Eretmapodites* in brown. Scale bar at 0.05 is shown.

The online version of this article includes the following source data and figure supplement(s) for figure 4:

**Source data 1.** Multiple sequence alignment of 122 28S rRNA sequences, including two sequences from

*Figure 4 continued on next page*

*Figure 4 continued*

*Horreolanus* sp. (FASTA).

**Source data 2.** Multiple sequence alignment of 114 18S rRNA sequences, including two sequences from *Horreolanus* sp. (FASTA).

**Figure supplement 1.** Phylogenetic tree based on 28S rRNA sequences generated from this study (3,900 bp).

**Figure supplement 2.** Phylogenetic tree based on 18S rRNA sequences (1,900 bp).

## rRNA as a molecular marker for taxonomy and phylogeny

We sequenced a 621 bp region of the COI gene to confirm morphological species identification of our specimens and to compare the functionality of rRNA and COI sequences as molecular markers for taxonomic and phylogenetic investigations. COI sequences were able to unequivocally determine the species identity in most specimens except for the following cases. *An. coustani* COI sequences from our study, regardless of specimen origin, shared remarkably high nucleotide similarity (>98%) with several other *Anopheles* species such as *An. rhodesiensis*, *An. rufipes*, *An. ziemanni*, *An. tenebrosus*, although *An. coustani* remained the most frequent and closest match. In the case of *Ae. simpsoni*, three specimens had been morphologically identified as *Ae. opok* although their COI sequences showed 97–100% similarity to that of *Ae. simpsoni*. As GenBank held no records of *Ae. opok* COI at the time of this study, we instead aligned the putative *Ae. simpsoni* COI sequences against two sister species of *Ae. opok*: *Ae. luteocephalus* and *Ae. africanus*. We found they shared only 90% and 89% similarity, respectively. Given this significant divergence, we concluded these specimens to be *Ae. simpsoni*. Ambiguous results were especially frequent among *Culex* specimens belonging to the *Cx. pipiens* or *Cx. vishnui* subgroups, where the query sequence differed with either of the top two hits by a single nucleotide. For example, between *Cx. quinquefasciatus* and *Cx. pipiens* of the *Cx. pipiens* subgroup, and between *Cx. vishnui* and *Cx. tritaeniorhynchus* of the *Cx. vishnui* subgroup.

Among our three specimens of *Ma. titillans*, two appeared to belong to a single species that is different from but closely related to *Ma. titillans*. We surmised that these specimens could instead be *Ma. pseudotitillans* based on morphological similarity but were not able to verify this by molecular means as no COI reference sequence is available for this species. These specimens are hence putatively labelled as 'Ma sp.4'.

Phylogenetic reconstruction based on the COI sequences showed clustering of all species taxa into distinct clades, underlining the utility of the COI gene in molecular taxonomy (*Figure 5*; *Hebert et al., 2003*; *Ratnasingham and Hebert, 2007*). However, species delineation among members of *Culex* subgroups were not as clear-cut, although sister species were correctly placed as sister taxa (*Figure 5*, in coral). This is comparable to the 28S+18S rRNA tree (*Figure 4*, in coral) and is indicative of lower intraspecies distances relative to interspecies distances.

To evaluate the utility of 28S and 18S rRNA sequences for molecular taxonomy, we used the 28S+18S rRNA tree to discern the identity of six specimens for which COI sequencing could not be performed. These specimens include three unknown *Mansonia* species (Specimen ID S33–S35), a *Ma. uniformis* (Specimen ID S51), an *An. gambiae* (Specimen ID S47), and a *Ur. geometrica* (Specimen ID S113) (*Appendix 1—table 1*). Their positions in the 28S+18S rRNA tree relative to adjacent taxa confirms the morphological identification of all six specimens to the genus level and, for three of them, to the species level (*Figure 4*; *Mansonia* in dark green, *Anopheles* in purple, *Uranotaenia* in pink).

The phylogenetic relationships indicated by the COI tree compared to the 28S+18S rRNA tree present only few points of similarity, with key differences summarised in *Table 2*. COI-based phylogenetic inference indeed showed clustering of generic taxa into monophyletic clades albeit with very weak bootstrap support, except for genera *Culex* and *Mansonia* (*Figure 5*; *Culex* in coral, *Mansonia* in dark green). Contrary to the 28S+18S rRNA tree (*Figure 4*), *Culex* subgenus *Melanoconion* was depicted as a polyphyletic taxon with *Cx. spissipes* being a part of the greater *Culicini* clade with members from subgenera *Oculeomyia* and *Culex* while *Cx. pedroi* and *Cx. portesi* formed a distantly related clade. Among the *Mansonia* specimens, the two unknown Ma sp.4 specimens were not positioned as the nearest neighbours of *Ma. titillans* and instead appeared to have diverged earlier from most of the other taxa from the *Culicidae* family. Notably, the COI sequences of genus *Anopheles* (*Figure 5*, in purple) is not basal to the other members of *Culicidae* and is instead shown to be sister to *Culex* COI sequences (8% bootstrap support). This is a direct contrast to what is suggested by the

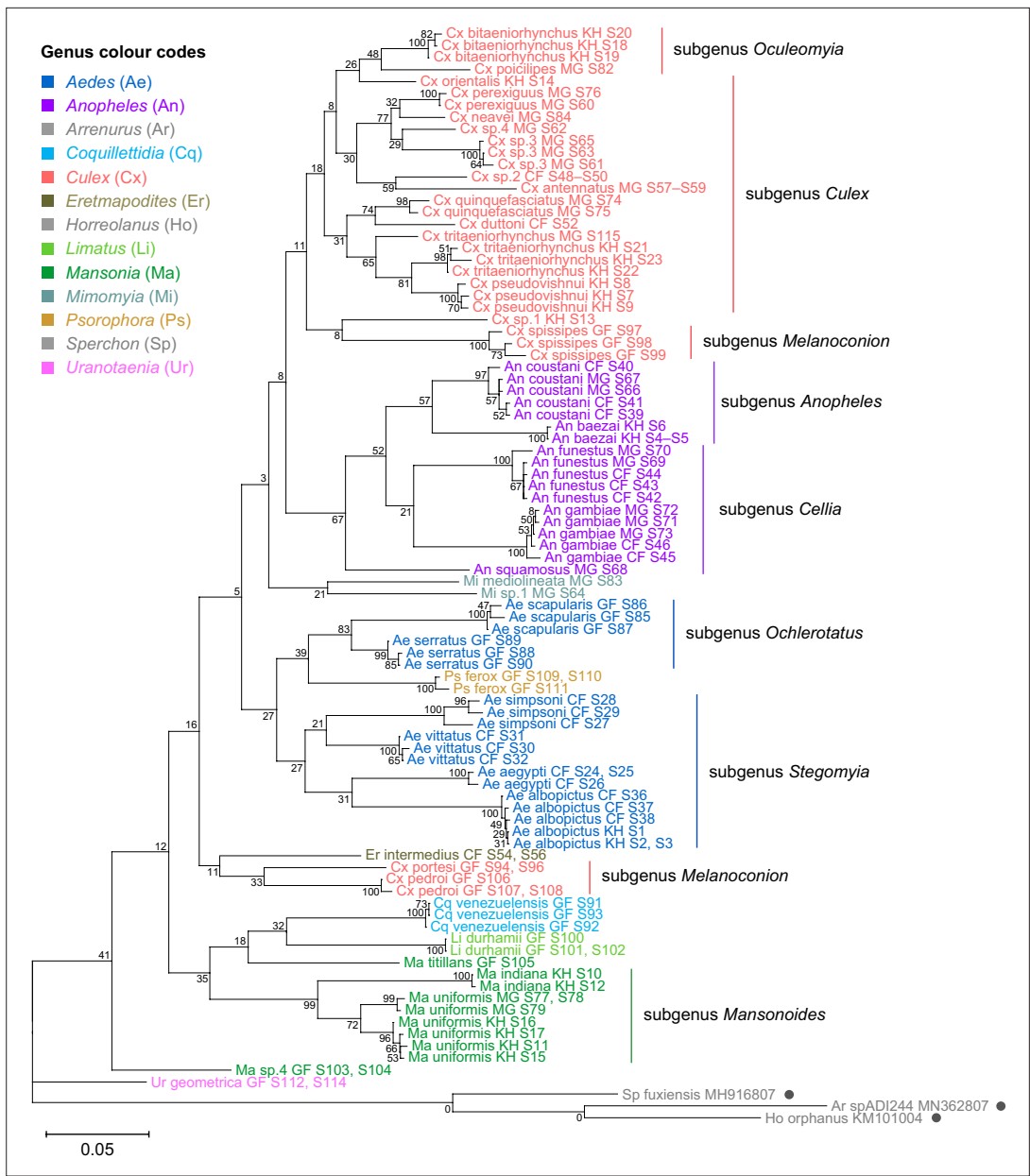

**Figure 5.** Cytochrome *c* oxidase I (COI) sequences cluster by species but show phylogenetic relationships that contrast those derived from rRNA trees. A phylogenetic tree based on COI sequences (621–699 bp) was inferred using the maximum-likelihood method and constructed to scale using MEGA X (***Kumar et al., 2018***) with three water mite species to serve as outgroups. Outgroup sequences obtained from GenBank are annotated with filled circles and their accession numbers are shown. Values at each node indicate bootstrap support (%) from 500 replications. Each specimen label contains information on taxonomy, origin (as indicated in two-letter country codes), and specimen ID. Specimen genera are indicted by colour: *Culex* in coral, *Anopheles* in purple, *Aedes* in dark blue, *Mansonia* in dark green, *Limatus* in light green, *Coquillettidia* in light blue, *Psorophora* in yellow, *Mimomyia* in teal, *Uranotaenia* in pink, and *Eretmapodites* in brown. Scale bar at 0.05 is shown.

The online version of this article includes the following source data for figure 5:

**Source data 1.** Multiple sequence alignment of 106 cytochrome *c* oxidase I (COI) sequences (FASTA).

rRNA phylogenies (***Figures 3 and 4***, ***Figure 4—figure supplements 1 and 2***; *Anopheles* in purple), which suggests *Culex* (in coral) rRNA sequences to be among the most recently diverged. Bootstrap support for the more internal nodes of the COI trees were remarkably low compared to those of rRNA-based trees.

**Table 2.** Summary of differences between rRNA and cytochrome *c* oxidase I (COI) phylogenies.

| Taxa | 28S+18S rRNA phylogeny (Figure 4) | COI phylogeny (Figure 5) |
|---|---|---|
| The *Anopheles* genus | Forms a clade that is basal to the all other members of family Culicidae; interspecies branch lengths are notably long | Forms a sister clade to the *Culex* genus, and is depicted to have diverged more recently; interspecies branch lengths are comparable to that of other genera |
| The *Ur. geometrica* species | Forms a clade within the Culicinae subfamily lineage | Forms a clade that is basal to the all other members of family Culicidae |
| The Aedini tribe | Forms a monophyletic clade comprising the genera *Aedes, Eretmapodites,* and *Psorophora,* with the latter being an early divergent lineage | Does not form a monophyletic clade; the *Psorophora* clade is placed among *Aedes* taxa and the *Eretmapodites* clade is sister to a *Culex* subgenus *Melanoconion* clade |
| The *Culex* genus | Splits into two monophyletic clades with the three French Guyanese species forming a closely related minor clade | Splits into two clades with two out of three French Guyanese species (*Cx. pedroi* and *Cx. portesi*) forming a distantly related minor clade, while the third (*Cx. spissipes*) is a part of the greater clade |
| The *Mansonia* genus | Is a polyphyletic group comprising two clades with the two French Guyanese taxa forming a distantly related minor clade; the major clade is placed among *Culex* taxa | Forms a subgenus *Mansonoides* clade as per the 28S+18S rRNA tree but the French Guyanese taxa do not cluster together; is depicted to have diverged earlier relative to other taxa in the assemblage |
| The Ma sp.4 species | Forms a sister clade to *Ma. titillans* as part of a minor French Guyanese *Mansonia* clade | Does not form a sister clade to *Ma. titillans;* instead is shown to have diverged earlier than all other members of family Culicidae after *Ur. geometrica* |

In all rRNA trees, it is clear that the interspecific and intersubgeneric evolutionary distances within the genus *Anopheles* are high relative to any other genera, indicating a greater degree of divergence (*Figure 3*, *Figure 3—figure supplement 1*, *Figure 4*, *Figure 4—figure supplements 1 and 2*; *Anopheles* in purple). This is evidenced by the longer branch lengths connecting Anopheline species-clades to the node of the most recent common ancestor for subgenera *Anopheles* and *Cellia*. This feature is not evident in the COI tree, where the Anopheline interspecies distances are comparable to those within the *Culex*, *Aedes*, and *Mansonia* taxa (*Figure 5*; *Anopheles* in purple, *Culex* in coral, *Aedes* in dark blue, *Mansonia* in dark green).

## On *Culex* subgroups

*Culex* (subgenus *Culex*) specimens of this study comprise several closely related sister species belonging to the *Cx. vishnui* and *Culex univittatus* subgroups, which are notoriously difficult to differentiate based on morphology. Accordingly, in the 28S+18S rRNA (*Figure 4*, in coral) and COI (*Figure 5*, in coral) trees these species and their known sister species were clustered together within the *Culex* (subgenus *Culex*) clade: *Cx. tritaeniorhynchus* with *Cx. pseudovishnui* (*Cx. vishnui* subgroup); *Cx. perexiguus* with *Cx. neavei* (*Cx. univittatus* subgroup).

The use of the COI sequence to distinguish between members of the *Culex* subgroups was limited. For example, for the two *Cx. quinquefasciatus* samples in our taxonomic assemblage (Specimen ID S74 and S75) (*Appendix 1—table 1*), BLAST analyses of their COI sequences revealed they are a single nucleotide apart from *Cx. pipiens* or *Cx. quinquefasciatus* COI sequences (*Appendix 2—table 1*). In the 28S rRNA tree with GenBank sequences (*Figure 3*), two *Cx. pipiens* GenBank sequences formed a clade sister to another containing three *Cx. quinquefasciatus* GenBank sequences and the 'Cx quinquefasciatus MG S74' sequence with 78% bootstrap support. This is in accordance with other studies examining mitochondrial sequences (*Sun et al., 2019*) and morphological attributes (*Harbach et al., 2017*). This shows that the 28S rRNA sequence can distinguish the two species and confirms that 'Cx quinquefasciatus MG S74' is indeed a *Cx. quinquefasciatus* specimen. However, 'Cx quinquefasciatus MG S75' is shown to be basal from other sequences within this *Cx. pipiens* subgroup-clade with 100% bootstrap support. Given that *Cx. quinquefasciatus* and *Cx. pipiens* are known to interbreed, it is plausible that this individual is a hybrid of the two species (*Farajollahi et al., 2011*).

## Discussion

RNA-seq metagenomics on field-captured sylvatic mosquitoes is a valuable tool for tracking mosquito viruses through surveillance and virus discovery. However, the lack of reference rRNA sequences hinders good oligo-based depletion and efficient clean-up of RNA-seq data. Additionally, de novo

assembly of rRNA sequences is complicated due to regions that are highly conserved across all distantly related organisms that could be present in a single specimen, that is, microbiota, parasites, or vertebrate blood meal. Hence, we established a method to bioinformatically filter out non-host rRNA reads for the accurate assembly of novel 28S and 18S rRNA reference sequences.

We found that phylogenetic reconstructions based on 28S sequences or concatenated 28S+18S rRNA sequences were able to correctly cluster mosquito taxa according to species and corroborate current mosquito classification. This demonstrates that our bioinformatics methodology reliably generates bona fide 28S and 18S rRNA sequences, even in specimens parasitized by water mites or engorged with vertebrate blood. Further, we were able to use 28S+18S rRNA sequence taxonomy for molecular species identification when COI sequences were unavailable or ambiguous, thus supporting the use of rRNA sequences as a molecular marker. In RNA-seq metagenomics applications, they have the advantage of circumventing the need to additionally isolate and sequence DNA from specimens, as RNA-seq reads can be directly mapped against reference sequences. In our hands, there are sufficient numbers of remaining reads post-depletion (5–10% of reads per sample) to assemble complete rRNA contigs (unpublished data).

Phylogenetic inferences based on 28S or 18S rRNA sequences alone do not recover the same interspecific relationships (*Figure 4—figure supplements 1 and 2*). Relative to 28S sequences, we observed more instances where multiple specimens have near-identical 18S rRNA sequences. This can occur for specimens belonging to the same species, but also for conspecifics sampled from different geographic locations, such as *An. coustani*, *An. gambiae*, or *Ae. albopictus*. More rarely, specimens from the same species subgroup, such as *Cx. pseudovishnui* and *Cx. tritaeniorhynchus*, also shared 18S rRNA sequences. This was surprising given that the 18S rRNA sequences in our dataset is 1,900 bp long. Concatenation of 28S and 18S rRNA sequences resolved this issue, enabling species delineation even among sister species of *Culex* subgroups, where morphological identification meets its limits.

In Cambodia and other parts of Asia, the *Cx. vishnui* subgroup includes *Cx. tritaeniorhynchus*, *Cx. vishnui*, and *Cx. pseudovishnui*, which are important vectors of JEV (*Maquart and Boyer, 2022*). The former two were morphologically identified in our study but later revealed by COI sequencing to be a sister species. Discerning sister species of the *Cx. pipiens* subgroup is further complicated by interspecific breeding, with some populations showing genetic introgression to varying extents (*Cornel et al., 2003*). The seven sister species of this subgroup are practically indistinguishable based on morphology and require molecular methods to discern (*Farajollahi et al., 2011*; *Zittra et al., 2016*). Indeed, the 621 bp COI sequence amplified in our study did not contain enough nucleotide divergence to allow clear identification, given that the COI sequence of *Cx. quinquefasciatus* specimens differed from that of *Cx. pipiens* by a single nucleotide. *Batovska et al., 2017*, found that even the Internal Transcribed Spacer 2 (ITS2) rDNA region, another common molecular marker, could not differentiate the two species. Other DNA molecular markers such as nuclear Ace-2 or CQ11 genes (*Aspen and Savage, 2003*; *Zittra et al., 2016*) or *Wolbachia pipientis* infection status (*Cornel et al., 2003*) are typically employed in tandem. In our study, 28S rRNA sequence-based phylogeny validated the identity of specimen 'Cx quinquefasciatus MG S74' (*Figure 3*, in coral) and suggested that specimen 'Cx quinquefasciatus MG S75' might have been a *pipiens-quinquefasciatus* hybrid. These examples demonstrate how 28S rRNA sequences, concatenated with 18S rRNA sequences or alone, contain enough resolution to differentiate between *Cx. pipiens* and *Cx. quinquefasciatus*. rRNA-based phylogeny thus allows for more accurate species identification and ecological observations in the context of disease transmission. Additionally, tracing the genetic flow across hybrid populations within the *Cx. pipiens* subgroup can inform estimates of vectorial capacity for each species. As only one or two members from the *Cx. pipiens* and *Cx. vishnui* subgroups were represented in our taxonomic assemblage, an explicit investigation including all member species of these subgroups in greater sample numbers is warranted to further test the degree of accuracy with which 28S and 18S rRNA sequences can delineate sister species.

Our study included French Guianese *Culex* species *Cx. spissipes* (group Spissipes), *Cx. pedroi* (group Pedroi), and *Cx. portesi* (group Vomerifer). These species belong to the New World subgenus *Melanoconion*, section Spissipes, with well-documented distribution in North and South Americas (*Sirivanakarn, 1982*) and are vectors of encephalitic alphaviruses EEEV and VEEV among others (*Talaga et al., 2021*; *Turell et al., 2008*; *Weaver et al., 2004*). Indeed, our rooted rRNA and COI trees showed the divergence of the three *Melanoconion* species from the major *Culex* clade comprising species broadly

found across Africa and Asia (*Auerswald et al., 2021*; *Farajollahi et al., 2011*; *Nchoutpouen et al., 2019*; *Takhampunya et al., 2011*). The topology of the concatenated 28S+18S rRNA tree places the *Cx. portesi* and *Cx. pedroi* species-clades as sister groups (92% bootstrap support), with *Cx. spissipes* as a basal group within the *Melanoconion* clade (100% bootstrap support) (*Figure 4*, in coral). This corroborates the systematics elucidated by *Navarro and Weaver, 2004*, using the ITS2 marker, and those by *Sirivanakarn, 1982* and *Sallum and Forattini, 1996* based on morphology. Curiously, in the COI tree, *Cx. spissipes* sequences were clustered with unknown species Cx. sp.1, forming a clade sister to another containing other *Culex* (*Culex*) and *Culex* (*Oculeomyia*) species, albeit with very low bootstrap support (*Figure 5*, in coral). Previous phylogenetic studies based on the COI gene have consistently placed *Cx. spissipes* or the Spissipes group basal to other groups within the *Melanoconion* subgenus (*Torres-Gutierrez et al., 2016*; *Torres-Gutierrez et al., 2018*). However, these studies contain only *Culex* (*Melanoconion*) species in their assemblage, apart from *Cx. quinquefasciatus* to act as an outgroup. This clustering of *Cx. spissipes* with non-*Melanoconion* species in our COI phylogeny could be an artefact of a much more diversified assemblage rather than a true phylogenetic link.

Taking advantage of our multi-country sampling, we examined whether rRNA or COI phylogeny can be used to distinguish conspecifics originating from different geographies. Our assemblage contains five of such species: *An. coustani*, *An. funestus*, *An. gambiae*, *Ae. albopictus*, and *Ma. uniformis*. Among the rRNA trees, the concatenated 28S+18S and 28S rRNA trees were able to discriminate between *Ma. uniformis* specimens from Madagascar, Cambodia, and the Central African Republic (in dark green), and between *An. coustani* specimens from Madagascar and the Central African Republic (in purple) (100% bootstrap support). In the COI tree, only *Ma. uniformis* was resolved into geographical clades comprising specimens from Madagascar and specimens from Cambodia (in dark green) (72% bootstrap support). No COI sequence was obtained from one *Ma. uniformis* specimen from the Central African Republic. The 28S+18S rRNA sequences ostensibly provided more population-level genetic information than COI sequences alone with better support. The use of rRNA sequences in investigating the biodiversity of mosquitoes should therefore be explored with a more comprehensive taxonomic assemblage.

The phylogenetic reconstructions based on rRNA or COI sequences in our study are hardly congruent (*Table 2*), but two principal differences stand out. First, the COI phylogeny does not recapitulate the early divergence of Anophelinae from Culicinae (*Figure 5*). This is at odds with other studies estimating mosquito divergence times based on mitochondrial genes (*Logue et al., 2013*; *Lorenz et al., 2021*) or nuclear genes (*Reidenbach et al., 2009*). The second notable feature in the rRNA trees is the remarkably large interspecies and intersubgeneric evolutionary distances within genus *Anopheles* relative to other genera in the Culicinae subfamily (*Figure 3*, *Figure 3—figure supplement 1*, *Figure 4*, *Figure 4—figure supplements 1 and 2*; *Anopheles* in purple) but this is not apparent in the COI tree. The hyperdiversity among *Anopheles* taxa may be attributed to the earlier diversification of the Anophelinae subfamily in the early Cretaceous period compared to that of the Culicinae subfamily—a difference of at least 40 million years (*Lorenz et al., 2021*). The differences in rRNA and COI tree topologies indicate a limitation in using COI alone to determine evolutionary relationships. Importantly, drawing phylogenetic conclusions from short DNA markers such as COI has been cautioned against due to its weak phylogenetic signal (*Hajibabaei et al., 2006*). The relatively short length of our COI sequences (621–699 bp) combined with the 100-fold higher nuclear substitution rate of mitochondrial genomes relative to nuclear genomes (*Arctander, 1995*) could result in homoplasy (*Danforth et al., 2005*), making it difficult to clearly discern ancestral sequences and correctly assign branches into lineages, as evidenced by the poor nodal bootstrap support at genus-level branches. Indeed, in the study by *Lorenz et al., 2021*, a phylogenetic tree constructed using a concatenation of all 13 protein-coding genes of the mitochondrial genome was able to resolve ancient divergence events. This affirms that while COI sequences can be used to reveal recent speciation events, longer or multi-gene molecular markers are necessary for studies into deeper evolutionary relationships (*Danforth et al., 2005*).

In contrast to Anophelines where 28S rRNA phylogenies illustrated higher interspecies divergence compared to COI phylogeny, two specimens of an unknown *Mansonia* species, 'Ma sp.4 GF S103' and 'Ma sp.4 GF S104', provided an example where interspecies relatedness based on their COI sequences is greater than that based on their rRNA sequences in relation to 'Ma titillans GF S105'.

While all rRNA trees placed 'Ma titillans GF S105' as a sister taxon with 100% bootstrap support, the COI tree placed M sp.4 basal to all other species except *Ur. geometrica* (*Figure 5*; *Mansonia* in dark green, *Uranotaenia* in pink). This may hint at a historical selective sweep in the mitochondrial genome, whether arising from geographical separation, mutations, or linkage disequilibrium with inherited symbionts (*Hurst and Jiggins, 2005*), resulting in the disparate mitochondrial haplogroups found in French Guyanese Ma sp.4 and *Ma. titillans*. In addition, both haplogroups are distant from those associated with members of subgenus *Mansonoides*. To note, the COI sequences of 'M sp.4 GF S103' and 'M sp.4 GF S104' share 87.12% and 87.39% nucleotide similarity, respectively, to that of 'Ma titillans GF S105'. Interestingly, the endosymbiont *Wo. pipientis* has been detected in *Ma. titillans* sampled from Brazil (*de Oliveira et al., 2015*), which may contribute to the divergence of 'Ma titillans GF S105' COI sequence away from those of Ma sp.4. This highlights other caveats of using a mitochondrial DNA marker in determining evolutionary relationships (*Hurst and Jiggins, 2005*), which nuclear markers such as 28S and 18S rRNA sequences may be immune to.

## Conclusions

Total RNA-seq is a valuable tool for surveillance and virus discovery in sylvatic mosquitoes but it is impeded by the lack of full-length rRNA reference sequences. Here, we presented an rRNA sequence assembly strategy and a dataset of 234 newly generated mosquito 28S and 18S rRNA sequences. Our work has expanded the current mosquito rRNA reference library by providing, to our knowledge, the first full-length rRNA records for 30 species in public databases and paves the way for the assembly of many more. These novel rRNA sequences can improve mosquito metagenomics based on RNA-seq

**Table 3.** Comparison of 28S or concatenated 28S+18S rRNA and cytochrome *c* oxidase I (COI) sequences as molecular markers.

| 28S+18S rRNA | |
| --- | --- |
| **Advantages** | **Disadvantages** |
| <ul><li>In RNA-seq metagenomics studies, molecular taxonomy of specimens based on rRNA sequences can be done from RNA-seq data without additional sample preparation or sequencing.</li><li>28S rRNA and concatenated 28S+18S rRNA sequences can resolve the identity of specimens where COI sequences were ambiguous, particularly between members of species subgroups.</li><li>28S rRNA and concatenated 28S+18S rRNA sequences can distinguish conspecifics from different geographies for certain species.</li><li>Phylogenetic inferences based on 28S rRNA and concatenated 28S+18S rRNA sequences show relationships that are more concordant to contemporary mosquito systematics elucidated by other studies and may be a more suitable marker to study deep evolutionary relationships.</li><li>Being longer and nuclear-encoded, 28S or concatenated 28S+18S rRNA sequences are immune to homoplasy or to selective sweeps that may affect genomes of inherited symbionts such as mitochondria.</li></ul> | <ul><li>RNA-seq costs more than Sanger sequencing.</li><li>Reference rRNA sequences are currently much more limited in breadth compared to other established molecular markers.</li></ul> |

| COI | |
| --- | --- |
| **Advantages** | **Disadvantages** |
| <ul><li>With a larger reference database, the COI is a versatile marker for molecular taxonomy.</li><li>Being a shorter DNA marker, the COI gene is cost- and time-effective to amplify, sequence, and characterise.</li><li>Universal primer sets to amplify the COI marker have been developed and tested for many diverse species.</li></ul> | <ul><li>All species taxa clustered into distinct clades but with weaker bootstrap support at internal nodes relative to those of the 28S+18S rRNA tree.</li><li>For *An. coustani*, and members of *Culex* species subgroups such as *Cx. quinquefasciatus* and *Cx. tritaeniorhynchus*, COI sequences are unable to unequivocally confirm species identity as species can differ by just one nucleotide. Other molecular markers are often used in tandem.</li></ul> |

by enabling physical and computational removal of rRNA from specimens and streamlined species identification using rRNA markers.

Given that a reference sequence is available, rRNA markers could serve as a better approach for mosquito taxonomy and phylogeny than COI markers. In analysing the same set of specimens based on their COI and rRNA sequences, we showed that rRNA sequences can discriminate between members of a species subgroup as well as conspecifics from different geographies. Phylogenetic inferences from a tree based on 28S rRNA sequences alone or on concatenated 28S+18S rRNA sequences are more aligned with contemporary mosquito systematics, showing evolutionary relationships that agree with other phylogenetic studies. While COI-based phylogeny can reveal recent speciation events, rRNA sequences may be better suited for investigations of deeper evolutionary relationships as they are less prone to selective sweeps and homoplasy. The advantages and disadvantages of rRNA and COI sequences as molecular markers are summarised in *Table 3*. Further studies are necessary to reveal how rRNA sequences compare against other nuclear or mitochondrial DNA marker systems (*Batovska et al., 2017*; *Beebe, 2018*; *Behura, 2006*; *Ratnasingham and Hebert, 2007*; *Reidenbach et al., 2009*; *Vezenegho et al., 2022*).

## Materials and methods
### Sample collection
Mosquito specimens were sampled from 2019 to 2020 by medical entomology teams from the Institut Pasteur de Bangui (Central African Republic, Africa; CF), Institut Pasteur de Madagascar (Madagascar, Africa; MG), Institut Pasteur du Cambodge (Cambodia, Asia; KH), and Institut Pasteur de la Guyane (French Guiana, South America; GF). Adult mosquitoes were sampled using several techniques including CDC light traps, BG sentinels, and human-landing catches. Sampling sites are sylvatic locations including rural settlements in the Central African Republic, Madagascar, and French Guiana and national parks in Cambodia. Mosquitoes were morphologically identified using taxonomic identification keys (*Edwards, 1941*; *Grjebine, 1966*; *Huang and Ward, 1981*; *Oo et al., 2006*; *Rattanarithikul et al., 2007*; *Rattanarithikul et al., 2010*; *Rattanarithikul et al., 2005a*; *Rattanarithikul et al., 2005b*; *Rattanarithikul et al., 2006a*; *Rattanarithikul et al., 2006b*; *Rueda, 2004*) on cold tables before preservation by flash freezing in liquid nitrogen and transportation in dry ice to Institut Pasteur Paris for analysis. A list of the 112 mosquito specimens included in our taxonomic assemblage and their related information are provided in *Appendix 1—table 1*. To note, specimen ID S53, S80, and S81 were removed from our assemblage as their species identity could not be determined by COI or rRNA sequences.

### RNA and DNA isolation
Nucleic acids were isolated from mosquito specimens using TRIzol reagent according to the manufacturer's protocol (Invitrogen, Thermo Fisher Scientific, Waltham, MA, USA). Single mosquitoes were homogenised into 200 µL of TRIzol reagent and other of the reagents within the protocol were volume-adjusted accordingly. Following phase separation, RNA were isolated from the aqueous phase while DNA were isolated from the remaining interphase and phenol-chloroform phase. From here, RNA is used to prepare cDNA libraries for next-generation sequencing while DNA is used in PCR amplification and Sanger sequencing of the mitochondrial COI gene as further described below.

### Probe depletion of rRNA
We tested a selective rRNA depletion protocol by *Morlan et al., 2012* on several mosquito species from the *Aedes*, *Culex*, and *Anopheles* genera. We designed 77 tiled 80 bp DNA probes antisense to the *Ae. aegypti* 28S, 18S, and 5.8S rRNA sequences. A pool of probes at a concentration of 0.04 µM were prepared. To bind probes to rRNA, 1 µL of probes and 2 µL of Hybridisation Buffer (100 mM Tris-HCl and 200 mM NaCl) were added to rRNA samples to a final volume of 20 µL and subjected to a slow-cool incubation starting at 95°C for 2 min, then cooling to 22°C at a rate of 0.1°C per second, ending with an additional 5 min at 22°C. The resulting RNA:DNA hybrids were treated with 2.5 µL Hybridase Thermostable RNase H (Epicentre, Illumina, Madison, WI, USA) and incubated at 37°C for 30 min. To remove DNA probes, the mix was treated with 1 µL DNase I (Invitrogen) and purified with

Agencourt RNAClean XP Beads (Beckman Coulter, Brea, CA, USA). The resulting RNA is used for total RNA-seq to check depletion efficiency.

## Total RNA-seq

To obtain rRNA sequences, RNA samples were quantified on a Qubit Fluorometer (Invitrogen) using the Qubit RNA BR Assay kit (Invitrogen) for concentration adjustment. Non-depleted total RNA was used for library preparation for next-generation sequencing using the NEBNext Ultra II RNA Library Preparation Kit for Illumina (New England Biolabs, Ipswich, MA, USA) and the NEBNext Multiplex Oligos for Illumina (Dual Index Primers Set 1) (New England Biolabs). Sequencing was performed on a NextSeq500 sequencing system (Illumina, San Diego, CA, USA). Quality control of fastq data and trimming of adapters were performed with FastQC and cutadapt, respectively.

## 28S and 18S rRNA assembly

To obtain 28S and 18S rRNA contigs, we had to first clean our fastq library by separating the reads representing mosquito rRNA from all other reads. To achieve this, we used the SILVA RNA sequence database to create two libraries: one containing all rRNA sequences recorded under the 'Insecta' node of the taxonomic tree, the other containing the rRNA sequences of many other nodes distributed throughout the taxonomic tree, hence named 'Non-Insecta' (*Quast et al., 2013*). Each read was aligned using the nucleotide Basic Local Alignment Search Tool (BLASTn, https://blast.ncbi.nlm.nih.gov/) of the National Center for Biotechnology Information (NCBI) against each of the two libraries and the scores of the best high-scoring segment pairs from the two BLASTns are subsequently used to calculate a ratio of Insecta over Non-Insecta scores (*Altschul et al., 1990*). Only reads with a ratio greater than 0.8 were used in the assembly. The two libraries being non-exhaustive, we chose this threshold of 0.8 to eliminate only reads that were clearly of a non-insect origin. Selected reads were assembled with the SPAdes genome assembler using the '-rna' option, allowing more heterogeneous coverage of contigs and kmer lengths of 31, 51, and 71 bases (*Bankevich et al., 2012*). This method successfully assembled rRNA sequences for all specimens, including a parasitic *Horreolanus* water mite (122 sequences for 28S and 114 sequences for 18S).

Initially, our filtration technique had two weaknesses. First, there is a relatively small number of complete rRNA sequences in the Insecta library from SILVA. To compensate for this, we carried out several filtration cycles, each time adding in the complete sequences produced in previous cycles to the Insecta library. Second, when our mosquito specimens were parasitized by other insects, it was not possible to bioinformatically filter out rRNA reads belonging to the parasite. For these rare cases, we used the '`--trusted-contigs`' option of the SPAdes assembler (*Bankevich et al., 2012*), giving it access to the 28S and 18S rRNA sequences of the mosquito closest in terms of taxonomic distance. By doing this, the assembler was able to reconstruct the rRNA of the mosquito as well as the rRNA of the parasitizing insect. All assembled rRNA sequences from this study have been deposited in GenBank with accession numbers OM350214–OM350327 for 18S rRNA sequences and OM542339–OM542460 for 28S rRNA sequences.

## COI amplicon sequencing

The mitochondrial COI gene was amplified from DNA samples using the universal 'Folmer' primer set LCO1490 (5'- GGTCAACAAATCATAAAGATATTGG -3') and HCO2198 (5'-TAAACTTCAGGGTGAC CAAAAAATCA-3'), as per standard COI marker sequencing practices, producing a 658 bp product (*Folmer et al., 1994*). PCRs were performed using Phusion High-Fidelity DNA Polymerase (Thermo Fisher Scientific). Every 50 µL reaction contained 10 µL of 5× High Fidelity buffer, 1 µL of 10 mM dNTPs, 2.5 µL each of 10 mM forward (LCO1490) and reverse (HCO2198) primer, 28.5 µL of water, 5 µL of DNA sample, and 0.5 µL of 2 U/µL Phusion DNA polymerase. A three-step cycling incubation protocol was used: 98°C for 30 s; 35 cycles of 98°C for 10 s, 60°C for 30 s, and 72°C for 15 s; 72°C for 5 min ending with a 4°C hold. PCR products were size-verified using gel electrophoresis and then gel-purified using the QIAquick Gel Extraction Kit (Qiagen, Hilden, Germany). Sanger sequencing of the COI amplicons were performed by Eurofins Genomics, Ebersberg, Germany.

## COI sequence analysis

Forward and reverse COI DNA sequences were end-trimmed to remove bases of poor quality (Q score <30). At the 5' ends, sequences were trimmed at the same positions such that all forward sequences start with 5'-TTTTGG and all reverse sequences start with 5'-GGNTCT. Forward and reverse sequences were aligned using BLAST to produce a 621 bp consensus sequence. In cases where good quality sequences extends beyond 621 bp, forward and reverse sequences were assembled using Pearl (https://www.gear-genomics.com/pearl/) and manually checked for errors against trace files (*Rausch et al., 2019*; *Rausch et al., 2020*). We successfully assembled a total of 106 COI sequences. All assembled COI sequences from this study have been deposited in GenBank with accession numbers OM630610–OM630715.

## COI validation of morphology-based species identification

We analysed assembled COI sequences with BLASTn against the nucleotide collection (nr/nt) database to confirm morphology-based species identification. BLAST analyses revealed 32 cases where top hits indicated a different species identity, taking <95% nucleotide sequence similarity as the threshold to delineate distinct species (*Appendix 2—table 1*). In these cases, the COI sequence of the specimen was then BLAST-aligned against a GenBank record representing the morphological species to verify that the revised identity is a closer match by a significant margin, that is, more than 2% nucleotide sequence similarity. All species names reported hereafter reflect identities determined by COI sequence except for cases where COI-based identities were ambiguous, in which case morphology-based identities were retained. In cases where matches were found within a single genus but of multiple species, specimens were indicated as an unknown member of their genus (e.g., *Culex* sp.). Information of the highest-scoring references for all specimens, including details of ambiguous BLASTn results, are recorded in *Appendix 2—table 1*.

Within our COI sequences, we found six unidentified *Culex* species (including two that matched to GenBank entries identified only to the genus level), four unidentified *Mansonia* species, and one unidentified *Mimomyia* species. For *An. baezai*, no existing GenBank records were found at the time this analysis was performed.

## Phylogenetic analysis

Multiple sequence alignment (MSA) were performed on assembled COI and rRNA sequences using the MUSCLE software (*Edgar, 2004*; *Madeira et al., 2019*). As shown in *Figure 3—figure supplement 2*, the 28S rRNA sequences contain many blocks of highly conserved nucleotides, which makes the result of multiple alignment particularly evident. We therefore did not test other alignment programs. The multiple alignment of the COI amplicons is even more evident since no gaps are necessary for this alignment.

Phylogenetic tree reconstructions were performed with the MEGA X software using the maximum-likelihood method (*Kumar et al., 2018*). Default parameters were used with bootstrapping with 500 replications to quantify confidence level in branches. For rRNA trees, sequences belonging to an unknown species of parasitic water mite (genus *Horreolanus*) found in our specimens served as an outgroup taxon. In addition, we created and analysed a separate dataset combining our 28S rRNA sequences and full-length 28S rRNA sequences from GenBank totalling 169 sequences from 58 species (12 subgenera). To serve as outgroups for the COI tree, we included sequences obtained from GenBank of three water mite species, *Horreolanus orphanus* (KM101004), *Sperchon fuxiensis* (MH916807), and *Arrenurus* sp. (MN362807).

## Acknowledgements

We thank members of the Saleh lab for valuable discussions and Dr Louis Lambrechts for critical reading of the manuscript. We especially thank all medical entomology staff of IP Bangui, IP Cambodge (Sony Yean, Kimly Heng, Kalyan Chhuoy, Sreynik Nhek, Moeun Chhum, Kimhuor Sour, and Pierre-Olivier Maquart), IP Madagascar, and IP Guyane for assistance in field missions, laboratory work, and logistics, and Inès Partouche from IP Paris for laboratory assistance. We are also grateful to Dr Catherine Dauga for advice on phylogenetic analyses, and to Amandine Guidez for providing a French Guiana-specific COI reference library. Finally, we thank our Reviewers, including Dr Leslie Vosshall and Dr

Katherine Young, and Editor Dr Sara Sawyer for constructive reviews and comments. This work was supported by the Defence Advanced Research Projects Agency PREEMPT program managed by Dr Rohit Chitale and Dr Kerri Dugan (Cooperative Agreement HR001118S0017) (the content of the information does not necessarily reflect the position or the policy of the US government, and no official endorsement should be inferred).

## Additional information

### Funding

| Funder | Grant reference number | Author |
|---|---|---|
| Defense Advanced Research Projects Agency | Cooperative Agreement HR001118S0017 | Maria-Carla Saleh |

The funders had no role in study design, data collection and interpretation, or the decision to submit the work for publication.

### Author contributions

Cassandra Koh, Conceptualization, Data curation, Formal analysis, Investigation, Visualization, Methodology, Writing – original draft, Project administration, Writing – review and editing; Lionel Frangeul, Conceptualization, Resources, Data curation, Formal analysis, Investigation, Visualization, Methodology, Writing – review and editing; Hervé Blanc, Conceptualization, Data curation, Investigation, Methodology; Carine Ngoagouni, Philippe Dussart, Nina Grau, Romain Girod, Resources, Writing – review and editing; Sébastien Boyer, Jean-Bernard Duchemin, Resources, Formal analysis, Writing – review and editing; Maria-Carla Saleh, Conceptualization, Supervision, Funding acquisition, Investigation, Project administration, Writing – review and editing

### Author ORCIDs

Cassandra Koh ![ORCID] http://orcid.org/0000-0003-2466-6731
Philippe Dussart ![ORCID] http://orcid.org/0000-0002-1931-3037
Maria-Carla Saleh ![ORCID] http://orcid.org/0000-0001-8593-4117

### Decision letter and Author response

Decision letter https://doi.org/10.7554/eLife.82762.sa1
Author response https://doi.org/10.7554/eLife.82762.sa2

## Additional files

### Supplementary files

• Transparent reporting form

### Data availability

Multiple sequence alignment files are included as source data files. All sequences generated in this study have been deposited in GenBank under the accession numbers OM350214–OM350327 for 18S rRNA sequences, OM542339–OM542460 for 28S rRNA sequences, and OM630610–OM630715 for COI sequences.

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

# Appendix 1

**Appendix 1—table 1.** Taxonomic and sampling information on mosquito specimens and associated accession numbers of their cytochrome *c* oxidase I (COI), 18S rRNA, and 28S rRNA sequences (XLSX).

| Sequence ID | Taxonomy [Genus (subgenus) species] | Origin | Collection site | Collection period | Blood engorged (Y/N) | Sample ID | COI accession number | 18S rRNA accession number | 28S rRNA accession number |
|---|---|---|---|---|---|---|---|---|---|
| Ae_albopictus_KH_S1 | *Aedes (Stegomyia) albopictus* | Cambodia | Rattanakiri | Dec 2019 | N | 1 | OM630613 | OM350214 | OM542460 |
| Ae_albopictus_KH_S2 | *Aedes (Stegomyia) albopictus* | Cambodia | Rattanakiri | Dec 2019 | N | 2 | OM630614 | OM350220 | OM542373 |
| Ae_albopictus_KH_S3 | *Aedes (Stegomyia) albopictus* | Cambodia | Rattanakiri | Dec 2019 | N | 3 | OM630615 | OM350316 | OM542374 |
| An_baezai_KH_S4 | *Anopheles (Anopheles) baezai* | Cambodia | Koh Kong | Mar 2019 | N | 4 | OM630631 | OM350327 | OM542357 |
| An_baezai_KH_S5 | *Anopheles (Anopheles) baezai* | Cambodia | Koh Kong | Mar 2019 | N | 5 | OM630632 | OM350233 | OM542440 |
| An_baezai_KH_S6 | *Anopheles (Anopheles) baezai* | Cambodia | Koh Kong | Mar 2019 | N | 6 | OM630633 | OM350234 | OM542358 |
| Cx_pseudovishnui_KH_S7 | *Culex (Culex) pseudovishnui* | Cambodia | Rattanakiri | Dec 2019 | N | 7 | OM630689 | OM350285 | OM542413 |
| Cx_pseudovishnui_KH_S8 | *Culex (Culex) pseudovishnui* | Cambodia | Rattanakiri | Dec 2019 | N | 8 | OM630690 | OM350286 | OM542414 |
| Cx_pseudovishnui_KH_S9 | *Culex (Culex) pseudovishnui* | Cambodia | Rattanakiri | Dec 2019 | N | 9 | OM630691 | OM350287 | OM542415 |
| Ma_indiana_KH_S10 | *Mansonia (Mansonioides) indiana* | Cambodia | Battambong | Nov 2019 | N | 10 | OM630698 | OM350295 | OM542422 |
| Ma_uniformis_KH_S11 | *Mansonia (Mansonioides) uniformis* | Cambodia | Battambong | Nov 2019 | N | 11 | OM630699 | OM350296 | OM542423 |
| Ma_indiana_KH_S12 | *Mansonia (Mansonioides) indiana* | Cambodia | Battambong | Nov 2019 | N | 12 | OM630700 | OM350297 | OM542424 |
| Cx_sp.1_KH_S13 | *Culex sp.1* | Cambodia | Prek Toal | Feb 2019 | N | 13 | OM630672 | OM350267 | OM542395 |
| Cx_orientalis_KH_S14 | *Culex (Culex) orientalis* | Cambodia | Prek Toal | Feb 2019 | N | 14 | OM630673 | OM350268 | OM542396 |
| Ma_uniformis_KH_S15 | *Mansonia (Mansonioides) uniformis* | Cambodia | Battambong | Nov 2019 | N | 15 | OM630705 | OM350303 | OM542430 |
| Ma_uniformis_KH_S16 | *Mansonia (Mansonioides) uniformis* | Cambodia | Battambong | Nov 2019 | N | 16 | OM630706 | OM350305 | OM542432 |
| Ma_uniformis_KH_S17 | *Mansonia (Mansonioides) uniformis* | Cambodia | Battambong | Nov 2019 | N | 17 | OM630707 | OM350304 | OM542431 |
| Cx_bitaeniorhynchus_KH_S18 | *Culex (Oculeomyia) bitaeniorhynchus* | Cambodia | Battambong | Nov 2019 | N | 18 | OM630656 | OM350255 | OM542381 |
| Cx_bitaeniorhynchus_KH_S19 | *Culex (Oculeomyia) bitaeniorhynchus* | Cambodia | Battambong | Nov 2019 | N | 19 | OM630657 | OM350256 | OM542382 |
| Cx_bitaeniorhynchus_KH_S20 | *Culex (Oculeomyia) bitaeniorhynchus* | Cambodia | Battambong | Nov 2019 | N | 20 | OM630658 | OM350257 | OM542383, OM542384 |
| Cx_tritaeniorhynchus_KH_S21 | *Culex (Culex) tritaeniorhynchus* | Cambodia | Battambong | Nov 2019 | N | 21 | OM630680 | OM350277 | OM542404 |

*Appendix 1—table 1 Continued on next page*

Appendix 1—table 1 Continued

| Sequence ID | Taxonomy [Genus (subgenus) species] | Origin | Collection site | Collection period | Blood engorged (Y/N) | Sample ID | COI accession number | 18S rRNA accession number | 28S rRNA accession number |
|---|---|---|---|---|---|---|---|---|---|
| Cx_tritaeniorhynchus_KH_S22 | *Culex* (*Culex*) *tritaeniorhynchus* | Cambodia | Battambong | Nov 2019 | N | 22 | OM630681 | OM350278 | OM542405 |
| Cx_tritaeniorhynchus_KH_S23 | *Culex* (*Culex*) *tritaeniorhynchus* | Cambodia | Battambong | Nov 2019 | N | 23 | OM630682 | OM350279 | OM542406 |
| Ae_aegypti_CF_S24 | *Aedes* (*Stegomyia*) *aegypti* | Central African Republic | Pissa | Jun 2019 | N | 24 | OM630610 | OM350314 | OM542339 |
| Ae_aegypti_CF_S25 | *Aedes* (*Stegomyia*) *aegypti* | Central African Republic | Pissa | Jun 2019 | N | 25 | OM630611 | OM350215 | OM542340 |
| Ae_aegypti_CF_S26 | *Aedes* (*Stegomyia*) *aegypti* | Central African Republic | Pissa | Jun 2019 | N | 26 | OM630612 | OM350216 | OM542341 |
| Ae_simpsoni_CF_S27 | *Aedes* (*Stegomyia*) *simpsoni* | Central African Republic | Pissa | Jun 2019 | N | 27 | OM630619 | OM350221 | OM542345 |
| Ae_simpsoni_CF_S28 | *Aedes* (*Stegomyia*) *simpsoni* | Central African Republic | Pissa | Jun 2019 | N | 28 | OM630620 | OM350222 | OM542346 |
| Ae_simpsoni_CF_S29 | *Aedes* (*Stegomyia*) *simpsoni* | Central African Republic | Pissa | Jun 2019 | N | 29 | OM630621 | OM350223 | OM542347 |
| Ae_vittatus_CF_S30 | *Aedes* (*Fredwardsius*) *vittatus* | Central African Republic | Gbozo | Aug 2019 | Y | 30 | OM630628 | OM350230 | OM542439 |
| Ae_vittatus_CF_S31 | *Aedes* (*Fredwardsius*) *vittatus* | Central African Republic | Gbozo | Aug 2019 | N | 31 | OM630629 | OM350231 | OM542355 |
| Ae_vittatus_CF_S32 | *Aedes* (*Fredwardsius*) *vittatus* | Central African Republic | Gbozo | Aug 2019 | N | 32 | OM630630 | OM350232 | OM542356 |
| Ma_sp.1_CF_S33 | *Mansonia* sp.1 | Central African Republic | Bayanga | Nov 2019 | Y | 33 | N/A | OM350294 | OM542449 |
| Ma_sp.2_CF_S34 | *Mansonia* sp.2 | Central African Republic | Bayanga | Nov 2019 | Y | 34 | N/A | OM350322 | OM542450, OM542456 |
| Ho_sp.1_CF_S34 | *Horreolanus* sp.1 | Central African Republic | Bayanga | Nov 2019 | – | 34 | N/A | OM350325 | OM542457 |
| Ho_sp.2_CF_S34 | *Horreolanus* sp.2 | Central African Republic | Bayanga | Nov 2019 | – | 34 | N/A | OM350326 | OM542458 |
| Ma_sp.3_CF_S35 | *Mansonia* sp.3 | Central African Republic | Bayanga | Nov 2019 | Y | 35 | N/A | OM350323 | OM542451 |
| Ae_albopictus_CF_S36 | *Aedes* (*Stegomyia*) *albopictus* | Central African Republic | Pissa | Jun 2019 | N | 36 | OM630616 | OM350217 | OM542342 |
| Ae_albopictus_CF_S37 | *Aedes* (*Stegomyia*) *albopictus* | Central African Republic | Pissa | Jun 2019 | N | 37 | OM630617 | OM350218 | OM542343 |
| Ae_albopictus_CF_S38 | *Aedes* (*Stegomyia*) *albopictus* | Central African Republic | Pissa | Jun 2019 | N | 38 | OM630618 | OM350219 | OM542344 |

Appendix 1—table 1 Continued

| Sequence ID | Taxonomy [Genus (subgenus) species] | Origin | Collection site | Collection period | Blood engorged (Y/N) | Sample ID | COI accession number | 18S rRNA accession number | 28S rRNA accession number |
|---|---|---|---|---|---|---|---|---|---|
| An_coustani_CF_S39 | *Anopheles* (*Anopheles*) *coustani* | Central African Republic | Pissa | Jan 2020 | N | 39 | OM630634 | OM350235 | OM542359 |
| An_coustani_CF_S40 | *Anopheles* (*Anopheles*) *coustani* | Central African Republic | Pissa | Jan 2020 | N | 40 | OM630635 | OM350236 | OM542360 |
| An_coustani_CF_S41 | *Anopheles* (*Anopheles*) *coustani* | Central African Republic | Pissa | Jan 2020 | N | 41 | OM630636 | OM350237 | OM542361 |
| An_funestus_CF_S42 | *Anopheles* (*Cellia*) *funestus* | Central African Republic | Pissa | Jun 2019 | Y | 42 | OM630640 | OM350241 | OM542365 |
| An_funestus_CF_S43 | *Anopheles* (*Cellia*) *funestus* | Central African Republic | Pissa | Jun 2019 | Y | 43 | OM630641 | OM350242 | OM542366 |
| An_funestus_CF_S44 | *Anopheles* (*Cellia*) *funestus* | Central African Republic | Pissa | Jun 2019 | Y | 44 | OM630642 | OM350243 | OM542367 |
| An_gambiae_CF_S45 | *Anopheles* (*Cellia*) *gambiae* | Central African Republic | Pissa | Jun 2019 | Y | 45 | OM630645 | OM350245 | OM542369, OM542370 |
| An_gambiae_CF_S46 | *Anopheles* (*Cellia*) *gambiae* | Central African Republic | Pissa | Jun 2019 | Y | 46 | OM630646 | OM350246 | OM542371 |
| An_gambiae_CF_S47 | *Anopheles* (*Cellia*) *gambiae* | Central African Republic | Pissa | Jun 2019 | Y | 47 | N/A | OM350247 | OM542372 |
| Cx_sp.2_CF_S48 | *Culex* sp.2 | Central African Republic | Bayanga | Nov 2019 | Y | 48 | OM630669 | OM350269 | OM542446 |
| Cx_sp.2_CF_S49 | *Culex* sp.2 | Central African Republic | Bayanga | Nov 2019 | Y | 49 | OM630670 | OM350315 | OM542397 |
| Cx_sp.2_CF_S50 | *Culex* sp.2 | Central African Republic | Bayanga | Nov 2019 | Y | 50 | OM630671 | OM350270 | OM542398 |
| Ma_uniformis_CF_S51 | *Mansonia* (*Mansonioides*) *uniformis* | Central African Republic | Bouar | May 2019 | Y | 51 | N/A | OM350301 | OM542428 |
| Cx_duttoni_CF_S52 | *Culex* (*Culex*) *duttoni* | Central African Republic | Mbaiki | Jan 2019 | Y | 52 | OM630704 | OM350302 | OM542429 |
| Er_intermedius_CF_S54 | *Eretmapodites intermedius* | Central African Republic | Pissa | Jun 2019 | N | 54 | OM630692 | OM350288 | OM542416 |
| Er_intermedius_CF_S55 | *Eretmapodites intermedius* | Central African Republic | Pissa | Jun 2019 | N | 55 | OM630693 | OM350289 | OM542417 |
| Er_intermedius_CF_S56 | *Eretmapodites intermedius* | Central African Republic | Pissa | Jun 2019 | N | 56 | OM630694 | OM350290 | OM542418 |
| Cx_antennatus_MG_S57 | *Culex* (*Culex*) *antennatus* | Madagascar | Ambato Boeny | Feb 2019 | N | 57 | OM630653 | OM350253 | OM542379 |
| Cx_antennatus_MG_S58 | *Culex* (*Culex*) *antennatus* | Madagascar | Ambato Boeny | Feb 2019 | N | 58 | OM630654 | OM350319 | OM542444 |

Appendix 1—table 1 Continued on next page

Appendix 1—table 1 Continued

| Sequence ID | Taxonomy [Genus (subgenus) species] | Origin | Collection site | Collection period | Blood engorged (Y/N) | Sample ID | COI accession number | 18S rRNA accession number | 28S rRNA accession number |
|---|---|---|---|---|---|---|---|---|---|
| Cx_antennatus_MG_S59 | *Culex (Culex) antennatus* | Madagascar | Ambato Boeny | Feb 2019 | N | 59 | OM630655 | OM350254 | OM542380 |
| Cx_perexiguus_MG_S60 | *Culex (Culex) perexiguus* | Madagascar | Amparafaravola | Feb 2019 | N | 60 | OM630660 | OM350258 | OM542386 |
| Cx_sp.3_MG_S61 | *Culex* sp.3 | Madagascar | Ambato Boeny | Aug 2019 | N | 61 | OM630661 | OM350259 | OM542387 |
| Cx_sp.4_MG_S62 | *Culex* sp.4 | Madagascar | Ambato Boeny | Aug 2019 | N | 62 | OM630662 | OM350260 | OM542388 |
| Cx_sp.3_MG_S63 | *Culex* sp.3 | Madagascar | Ambato Boeny | Feb 2019 | N | 63 | OM630686 | OM350282 | OM542410 |
| Mi_sp.1_MG_S64 | *Mimomyia* sp.1 | Madagascar | Ambato Boeny | Feb 2019 | N | 64 | OM630687 | OM350283 | OM542411 |
| Cx_sp.3_MG_S65 | *Culex* sp.3 | Madagascar | Ambato Boeny | Feb 2019 | N | 65 | OM630688 | OM350284 | OM542412 |
| An_coustani_MG_S66 | *Anopheles (Anopheles) coustani* | Madagascar | Ambato Boeny | Feb 2019 | N | 66 | OM630637 | OM350238 | OM542362 |
| An_coustani_MG_S67 | *Anopheles (Anopheles) coustani* | Madagascar | Ambato Boeny | Feb 2019 | N | 67 | OM630638 | OM350239 | OM542363 |
| An_squamosus_MG_S68 | *Anopheles (Cellia) squamosus* | Madagascar | Ambato Boeny | Feb 2019 | N | 68 | OM630639 | OM350240 | OM542364 |
| An_funestus_MG_S69 | *Anopheles (Cellia) funestus* | Madagascar | Ambato Boeny | Feb 2019 | N | 69 | OM630643 | OM350244 | OM542368 |
| An_funestus_MG_S70 | *Anopheles (Cellia) funestus* | Madagascar | Ambato Boeny | Feb 2020 | N | 70 | OM630644 | OM350317 | OM542441 |
| An_gambiae_MG_S71 | *Anopheles (Cellia) gambiae* | Madagascar | Ambato Boeny | Feb 2019 | N | 71 | OM630647 | OM350249 | OM542442 |
| An_gambiae_MG_S72 | *Anopheles (Cellia) gambiae* | Madagascar | Ambato Boeny | Feb 2019 | N | 72 | OM630648 | OM350248 | OM542443 |
| An_gambiae_MG_S73 | *Anopheles (Cellia) gambiae* | Madagascar | Ambato Boeny | Feb 2019 | N | 73 | OM630649 | OM350318 | OM542459 |
| Cx_quinquefasciatus_MG_S74 | *Culex (Culex) quinquefasciatus* | Madagascar | Amparafaravola | Feb 2019 | N | 74 | OM630674 | OM350271 | OM542399 |
| Cx_quinquefasciatus_MG_S75 | *Culex (Culex) quinquefasciatus* | Madagascar | Amparafaravola | Feb 2019 | N | 75 | OM630675 | OM350272 | OM542447 |
| Cx_perexiguus_MG_S76 | *Culex (Culex) perexiguus* | Madagascar | Mampikony | Aug 2019 | N | 76 | OM630676 | OM350273 | OM542400 |
| Ma_uniformis_MG_S77 | *Mansonia (Mansonioides) uniformis* | Madagascar | Ambato Boeny | Feb 2019 | N | 77 | OM630708 | OM350306 | OM542433 |
| Ma_uniformis_MG_S78 | *Mansonia (Mansonioides) uniformis* | Madagascar | Ambato Boeny | Feb 2019 | N | 78 | OM630709 | OM350307 | OM542434 |
| Ma_uniformis_MG_S79 | *Mansonia (Mansonioides) uniformis* | Madagascar | Ambato Boeny | Feb 2019 | N | 79 | OM630710 | OM350308 | OM542435 |
| Cx_poicilipes_MG_S82 | *Culex poicilipes* | Madagascar | Mampikony | Feb 2019 | N | 82 | OM630659 | OM350320 | OM542385, OM542445 |
| Mi_mediolineata_MG_S83 | *Mimomyia mediolineata* | Madagascar | Ambato Boeny | Feb 2019 | N | 83 | OM630683 | OM350280 | OM542407 |
| Cx_neavei_MG_S84 | *Culex (Culex) neavei* | Madagascar | Ambato Boeny | Feb 2019 | N | 84 | OM630684 | OM350281 | OM542408, OM542409 |
| Ae_scapularis_GF_S85 | *Aedes (Ochlerotatus) scapularis* | French Guiana | Hameau Prefontaine | Jul 2019 | N | 85 | OM630624 | OM350224 | OM542348, OM542349 |

Appendix 1—table 1 Continued on next page

Appendix 1—table 1 Continued

| Sequence ID | Taxonomy [Genus (subgenus) species] | Origin | Collection site | Collection period | Blood engorged (Y/N) | Sample ID | COI accession number | 18S rRNA accession number | 28S rRNA accession number |
|---|---|---|---|---|---|---|---|---|---|
| Ae_scapularis_GF_S86 | *Aedes (Ochlerotatus) scapularis* | French Guiana | Hameau Prefontaine | Jul 2019 | N | 86 | OM630622 | OM350225 | OM542350 |
| Ae_scapularis_GF_S87 | *Aedes (Ochlerotatus) scapularis* | French Guiana | Hameau Prefontaine | Jul 2019 | N | 87 | OM630623 | OM350226 | OM542351 |
| Ae_serratus_GF_S88 | *Aedes (Ochlerotatus) serratus* | French Guiana | Hameau Prefontaine | Nov 2020 | N | 88 | OM630625 | OM350227 | OM542352 |
| Ae_serratus_GF_S89 | *Aedes (Ochlerotatus) serratus* | French Guiana | Hameau Prefontaine | Nov 2020 | N | 89 | OM630626 | OM350228 | OM542353 |
| Ae_serratus_GF_S90 | *Aedes (Ochlerotatus) serratus* | French Guiana | Hameau Prefontaine | Nov 2020 | N | 90 | OM630627 | OM350229 | OM542354 |
| Cq_venezuelensis_GF_S91 | *Coquillettidia venezuelensis* | French Guiana | Hameau Prefontaine | Jul 2019 | N | 91 | OM630650 | OM350250 | OM542375 |
| Cq_venezuelensis_GF_S92 | *Coquillettidia venezuelensis* | French Guiana | Hameau Prefontaine | Jul 2019 | N | 92 | OM630651 | OM350251 | OM542376 |
| Cq_venezuelensis_GF_S93 | *Coquillettidia venezuelensis* | French Guiana | Hameau Prefontaine | Jul 2019 | N | 93 | OM630652 | OM350252 | OM542377, OM542378 |
| Cx_portesi_GF_S94 | *Culex* sp. BTLHVDV-2014 | French Guiana | Hameau Prefontaine | Jul 2019 | N | 94 | OM630666 | OM350264 | OM542392 |
| Cx_portesi_GF_S95 | *Culex* sp. BTLHVDV-2014 | French Guiana | Hameau Prefontaine | Jul 2019 | N | 95 | OM630667 | OM350265 | OM542393 |
| Cx_portesi_GF_S96 | *Culex* sp. BTLHVDV-2014 | French Guiana | Hameau Prefontaine | Jul 2019 | N | 96 | OM630668 | OM350266 | OM542394 |
| Cx_spissipes_GF_S97 | *Culex (Melanoconion)* sp. DJS-2020 | French Guiana | Hameau Prefontaine | Jul 2019 | N | 97 | OM630677 | OM350274 | OM542401 |
| Cx_spissipes_GF_S98 | *Culex (Melanoconion)* sp. DJS-2020 | French Guiana | Hameau Prefontaine | Jul 2019 | N | 98 | OM630678 | OM350275 | OM542402 |
| Cx_spissipes_GF_S99 | *Culex (Melanoconion)* sp. DJS-2020 | French Guiana | Hameau Prefontaine | Jul 2019 | N | 99 | OM630679 | OM350276 | OM542403 |
| Li_durhamii_GF_S100 | *Limatus durhamii* | French Guiana | Hameau Prefontaine | Jul 2019 | N | 100 | OM630695 | OM350291 | OM542419 |
| Li_durhamii_GF_S101 | *Limatus durhamii* | French Guiana | Hameau Prefontaine | Jul 2019 | N | 101 | OM630696 | OM350292 | OM542420 |
| Li_durhamii_GF_S102 | *Limatus durhamii* | French Guiana | Hameau Prefontaine | Jul 2019 | N | 102 | OM630697 | OM350293 | OM542421 |
| Ma_sp.4_GF_S103 | *Mansonia* sp.4 | French Guiana | Hameau Prefontaine | Jan 2020 | N | 103 | OM630701 | OM350298 | OM542425 |
| Ma_sp.4_GF_S104 | *Mansonia* sp.4 | French Guiana | Hameau Prefontaine | Jan 2020 | N | 104 | OM630702 | OM350299 | OM542426 |
| Ma_titillans_GF_S105 | *Mansonia (Mansonia) titillans* | French Guiana | Hameau Prefontaine | Jan 2020 | N | 105 | OM630703 | OM350300 | OM542427 |
| Cx_pedroi_GF_S106 | *Culex (Melanoconion) pedroi* | French Guiana | Hameau Prefontaine | Nov 2020 | N | 106 | OM630663 | OM350261 | OM542389 |

Appendix 1—table 1 Continued on next page

*Appendix 1—table 1 Continued*

| Sequence ID | Taxonomy [Genus (subgenus) species] | Origin | Collection site | Collection period | Blood engorged (Y/N) | Sample ID | COI accession number | 18S rRNA accession number | 28S rRNA accession number |
|---|---|---|---|---|---|---|---|---|---|
| Cx_pedroi_GF_S107 | *Culex (Melanoconion) pedroi* | French Guiana | Hameau Prefontaine | Nov 2020 | N | 107 | OM630664 | OM350262 | OM542390 |
| Cx_pedroi_GF_S108 | *Culex (Melanoconion) pedroi* | French Guiana | Hameau Prefontaine | Nov 2020 | N | 108 | OM630665 | OM350263 | OM542391 |
| Ps_ferox_GF_S109 | *Psorophora ferox* | French Guiana | Iracoubo | 2009 | N | 109 | OM630711 | OM350309 | OM542436 |
| Ps_ferox_GF_S110 | *Psorophora ferox* | French Guiana | Iracoubo | 2009 | N | 110 | OM630712 | OM350310 | OM542437 |
| Ps_ferox_GF_S111 | *Psorophora ferox* | French Guiana | Iracoubo | 2009 | N | 111 | OM630713 | OM350324 | OM542452 |
| Ur_geometrica_GF_S112 | *Uranotaenia (Uranotaenia) geometrica* | French Guiana | | 2010 | N | 112 | OM630714 | OM350311 | OM542453 |
| Ur_geometrica_GF_S113 | *Uranotaenia (Uranotaenia) geometrica* | French Guiana | | 2010 | N | 113 | N/A | OM350312 | OM542454 |
| Ur_geometrica_GF_S114 | *Uranotaenia (Uranotaenia) geometrica* | French Guiana | | 2010 | N | 114 | OM630715 | OM350313 | OM542438, OM542455 |
| Cx_tritaeniorhynchus_MG_S115 | *Culex (Culex) tritaeniorhynchus* | Madagascar | Ambato Boeny | Feb 2019 | N | 115 | OM630685 | OM350321 | OM542448 |

# Appendix 2

**Appendix 2—table 1.** Cytochrome *c* oxidase I (COI) sequence BLAST analyses summary (XLSX).

| Sequence ID | Sequence length | Morphological identification | BLASTn top hit species | BLASTn top hit accession | Query coverage | % identity | Comments |
|---|---|---|---|---|---|---|---|
| Ae_albopictus_KH_S1 | 699 | *Aedes albopictus* | *Aedes albopictus* | MK714006.1 | 99% | 99.71% | |
| Ae_albopictus_KH_S2 | 695 | *Aedes albopictus* | *Aedes albopictus* | MK714006.1 | 100% | 99.71% | |
| Ae_albopictus_KH_S3 | 695 | *Aedes albopictus* | *Aedes albopictus* | MK714006.1 | 100% | 99.71% | |
| An_baezai_KH_S4 | 658 | *Anopheles baezai* | *Anopheles darlingi* | MF381626.1 | 100% | 92.71% | *Anopheles baezai* not found in GenBank |
| An_baezai_KH_S5 | 670 | *Anopheles baezai* | *Anopheles darlingi* | MF381626.1 | 99% | 92.81% | *Anopheles baezai* not found in GenBank |
| An_baezai_KH_S6 | 659 | *Anopheles baezai* | *Anopheles darlingi* | MF381626.1 | 100% | 92.72% | *Anopheles baezai* not found in GenBank |
| Cx_pseudovishnui_KH_S7 | 660 | *Culex vishnui* | *Culex pseudovishnui* | MW321882.1 | 98% | 98.92% | 95% similarity to *Culex vishnui*, 94% similarity to *Culex tritaeniorhynchus* |
| Cx_pseudovishnui_KH_S8 | 659 | *Culex vishnui* | *Culex pseudovishnui* | MW321882.1 | 98% | 99.38% | 95% similarity to *Culex vishnui*, 94% similarity to *Culex tritaeniorhynchus* |
| Cx_pseudovishnui_KH_S9 | 659 | *Culex vishnui* | *Culex pseudovishnui* | MW321882.1 | 98% | 98.92% | 95% similarity to *Culex vishnui*, 94% similarity to *Culex tritaeniorhynchus* |
| Ma_indiana_KH_S10 | 660 | *Mansonia indiana* | *Mansonia indiana* | MK637632.1 | 98.00% | 99.54% | |
| Ma_uniformis_KH_S11 | 686 | *Mansonia indiana* | *Mansonia uniformis* | MK757484.1 | 99% | 99.71% | 89.99% similarity to *Mansonia indiana* MK637632.1 |
| Ma_indiana_KH_S12 | 693 | *Mansonia indiana* | *Mansonia indiana* | MK637632.1 | 97% | 99.41% | |
| Cx_sp.1_KH_S13 | 687 | *Culex quinquefasciatus* | *Culex (Lophoceraomyia)* sp.5 HY-2020 | MW321904.1 | 98% | 94.39% | 90% similarity to *Culex quinquefasciatus* GU188856.2 |
| Cx_orientalis_KH_S14 | 662 | *Culex quinquefasciatus* | *Culex orientalis* | MW228488.1 | 97% | 98.29% | |
| Ma_uniformis_KH_S15 | 658 | *Mansonia uniformis* | *Mansonia uniformis* | MK757484.1 | 100.00% | 99.54% | |
| Ma_uniformis_KH_S16 | 654 | *Mansonia uniformis* | *Mansonia uniformis* | MK757484.1 | 100.00% | 99.39% | |
| Ma_uniformis_KH_S17 | 657 | *Mansonia uniformis* | *Mansonia uniformis* | MK757484.1 | 99.00% | 99.54% | |
| Cx_bitaeniorhynchus_KH_S18 | 658 | *Culex bitaeniorhynchus* | *Culex bitaeniorhynchus* | HQ398898.1 | 97.00% | 99.69% | |
| Cx_bitaeniorhynchus_KH_S19 | 650 | *Culex bitaeniorhynchus* | *Culex bitaeniorhynchus* | HQ398898.1 | 98.00% | 99.84% | |
| Cx_bitaeniorhynchus_KH_S20 | 652 | *Culex bitaeniorhynchus* | *Culex bitaeniorhynchus* | HQ398898.1 | 98.00% | 99.38% | |
| Cx_tritaeniorhynchus_KH_S21 | 695 | *Culex tritaeniorhynchus* | *Culex vishnui* or *Culex tritaeniorhynchus* | MH374857.1 | 100% | 99.57% | 99.69% similarity to *Culex tritaeniorhynchus* MF179213.1 |
| Cx_tritaeniorhynchus_KH_S22 | 690 | *Culex tritaeniorhynchus* | *Culex vishnui* or *Culex tritaeniorhynchus* | MT876103.1 | 100% | 99.57% | |
| Cx_tritaeniorhynchus_KH_S23 | 663 | *Culex tritaeniorhynchus* | *Culex tritaeniorhynchus* | MT876103.1 | 99% | 98.79% | |

*Appendix 2—table 1 Continued on next page*

*Appendix 2—table 1 Continued*

| Sequence ID | Sequence length | Morphological identification | BLASTn top hit species | BLASTn top hit accession | Query coverage | % identity | Comments |
|---|---|---|---|---|---|---|---|
| Ae_aegypti_CF_S24 | 689 | *Aedes aegypti* | *Aedes aegypti* | MN299016.1 | 100% | 99.56% | |
| Ae_aegypti_CF_S25 | 660 | *Aedes aegypti* | *Aedes aegypti* | MN299024.1 | 100.00% | 99.70% | |
| Ae_aegypti_CF_S26 | 660 | *Aedes aegypti* | *Aedes aegypti* | MN299024.1 | 100.00% | 99.70% | |
| Ae_simpsoni_CF_S27 | 644 | *Aedes opok* | *Aedes simpsoni* | LC473669.1 | 97.00% | 97.77% | *Aedes opok* not found in GenBank, sequence has 90% and 89% similarity to *Aedes luteocephalus* and *Aedes africanus*, sister species of *Aedes opok*. |
| Ae_simpsoni_CF_S28 | 649 | *Aedes opok* | *Aedes simpsoni* | MN552302.1 | 99.00% | 100.00% | *Aedes. opok* not found in GenBank, sequence has 90% and 89% similarity to *Aedes luteocephalus* and *Aedes africanus*, sister species of *Aedes opok*. |
| Ae_simpsoni_CF_S29 | 627 | *Aedes opok* | *Aedes simpsoni* | MN552302.1 | 98.00% | 98.87% | *Aedes opok* not found in GenBank, sequence has 90% and 89% similarity to *Aedes luteocephalus* and *Aedes africanus*, sister species of *Aedes opok*. |
| Ae_vittatus_CF_S30 | 623 | *Aedes vittatus* | *Aedes vittatus* | MN552298.1 | 100.00% | 99.84% | |
| Ae_vittatus_CF_S31 | 622 | *Aedes vittatus* | *Aedes vittatus* | MN552298.1 | 100.00% | 99.68% | |
| Ae_vittatus_CF_S32 | 621 | *Aedes vittatus* | *Aedes vittatus* | MN552298.1 | 100.00% | 99.68% | |
| Ma_sp.1_CF_S33 | – | *Mansonia africana* | – | – | – | – | No COI obtained |
| Ma_sp.2_CF_S34 | – | *Mansonia africana* | – | – | – | – | No COI obtained |
| Ma_sp.3_CF_S35 | – | *Mansonia africana* | – | – | – | – | No COI obtained |
| Ae_albopictus_CF_S36 | 627 | *Aedes albopictus* | *Aedes albopictus* | MK995332.1 | 100.00% | 99.84% | |
| Ae_albopictus_CF_S37 | 621 | *Aedes albopictus* | *Aedes albopictus* | MK995332.1 | 100.00% | 100.00% | |
| Ae_albopictus_CF_S38 | 621 | *Aedes albopictus* | *Aedes albopictus* | MK995332.1 | 100.00% | 100.00% | |
| An_coustani_CF_S39 | 621 | *Anopheles coustani* | *Anopheles coustani* | MK585968.1 | 100.00% | 99.84% | |
| An_coustani_CF_S40 | 621 | *Anopheles coustani* | *Anopheles coustani* | MK585959.1 | 100.00% | 99.03% | |
| An_coustani_CF_S41 | 699 | *Anopheles coustani* | *Anopheles coustani* | MK585968.1 | 94.00% | 99.70% | |
| An_funestus_CF_S42 | 696 | *Anopheles funestus* | *Anopheles funestus* | MK300231.1 | 100.00% | 99.71% | |
| An_funestus_CF_S43 | 660 | *Anopheles funestus* | *Anopheles funestus* | MT375215.1 | 100.00% | 99.85% | |
| An_funestus_CF_S44 | 658 | *Anopheles funestus* | *Anopheles funestus* | MT375215.1 | 100.00% | 99.70% | |
| An_gambiae_CF_S45 | 660 | *Anopheles gambiae* | *Anopheles gambiae* | MG930895.1 | 86.00% | 99.79% | |
| An_gambiae_CF_S46 | 659 | *Anopheles gambiae* | *Anopheles gambiae* | MT375223.1 | 89.00% | 100.00% | |
| An_gambiae_CF_S47 | – | *Anopheles gambiae* | – | – | – | – | No COI obtained |
| Cx_sp.2_CF_S48 | 653 | *Culex quinquefasciatus* | *Culex corniger* | KM593015.1 | 100.00% | 94.95% | 94% similarity to all other *Culex* species |
| Cx_sp.2_CF_S49 | 660 | *Culex quinquefasciatus* | *Culex nigripalpus* | KM593058.1 | 99.00% | 94.65% | 94% similarity to all other *Culex* species |

*Appendix 2—table 1 Continued*

| Sequence ID | Sequence length | Morphological identification | BLASTn top hit species | BLASTn top hit accession | Query coverage | % identity | Comments |
|---|---|---|---|---|---|---|---|
| Cx_sp.2_CF_S50 | 658 | *Culex quinquefasciatus* | *Culex bidens* | MH931446.1 | 100.00% | 94.68% | 94% similarity to all other *Culex* species |
| Ma_uniformis_CF_S51 | – | *Mansonia uniformis* | – | – | – | – | No COI obtained |
| Cx_duttoni_CF_S52 | 621 | *Mansonia uniformis* | *Culex duttoni* | LC473629.1 | 100.00% | 99.68% | |
| Er_intermedius_CF_S54 | 620 | *Eretmapodites* sp. | *Eretmapodites intermedius* | MN552305.1 | 100.00% | 99.52% | |
| Er_intermedius_CF_S55 | 621 | *Eretmapodites* sp. | *Eretmapodites intermedius* | MN552305.1 | 100.00% | 99.68% | |
| Er_intermedius_CF_S56 | 621 | *Eretmapodites* sp. | *Eretmapodites intermedius* | MN552305.1 | 100.00% | 99.68% | |
| Cx_antennatus_MG_S57 | 621 | *Culex antennatus* | *Culex antennatus* | LC473659.1 | 100.00% | 100.00% | |
| Cx_antennatus_MG_S58 | 621 | *Culex antennatus* | *Culex antennatus* | LC473659.1 | 100.00% | 100.00% | |
| Cx_antennatus_MG_S59 | 621 | Culex. antennatus | *Culex antennatus* | LC473659.1 | 100.00% | 100.00% | |
| Cx_perexiguus_MG_S60 | 621 | *Culex decens* | *Culex perexiguus* | LC473634.1 | 100.00% | 99.84% | |
| Cx_sp.3_MG_S61 | 685 | *Culex decens* | Unknown *Culex* species | KU380436.1 | 96.00% | 96.05% | |
| Cx_sp.4_MG_S62 | 687 | *Culex decens* | Unknown *Culex* species | MT993494.1 | 99.00% | 95.63% | |
| Cx_sp.3_MG_S63 | 687 | *Culex univittatus* | Unknown *Culex* species | KU380436.1 | 95.00% | 96.50% | |
| Mi_sp.1_MG_S64 | 694 | *Culex univittatus* | *Mimomyia mimomyiaformis* | LC473719.1 | 94.00% | 92.55% | Unknown *Mimomyia* species |
| Cx_sp.3_MG_S65 | 691 | *Culex univittatus* | Unknown *Culex* species | KU380436.1 | 95.00% | 96.66% | |
| An_coustani_MG_S66 | 669 | *Anopheles coustani* | *Anopheles coustani* | NC_050693.1 | 99.00% | 99.40% | |
| An_coustani_MG_S67 | 659 | *Anopheles coustani* | *Anopheles coustani* | NC_050693.1 | 99.00% | 99.08% | |
| An_squamosus_MG_S68 | 653 | *Anopheles coustani* | *Anopheles squamosus* | MK776741.1 | 100.00% | 100.00% | |
| An_funestus_MG_S69 | 654 | *Anopheles funestus* | *Anopheles funestus* | MT375215.1 | 100.00% | 99.85% | |
| An_funestus_MG_S70 | 654 | *Anopheles funestus* | *Anopheles funestus* | MG742199.1 | 100.00% | 99.69% | |
| An_gambiae_MG_S71 | 654 | *Anopheles gambiae* | *Anopheles gambiae* | MT375222.1 | 100.00% | 99.85% | |
| An_gambiae_MG_S72 | 654 | *Anopheles gambiae* | *Anopheles gambiae* | MT375222.1 | 100.00% | 99.85% | |
| An_gambiae_MG_S73 | 622 | *Anopheles gambiae* | *Anopheles gambiae* | MT375222.1 | 100.00% | 100.00% | |
| Cx_quinquefasciatus_MG_S74 | 654 | *Culex quinquefasciatus* | *Culex pipiens* | MT199095.1 | 100.00% | 100.00% | 99.85% similarity to *Culex quinquefasciatus* |
| Cx_quinquefasciatus_MG_S75 | 647 | *Culex quinquefasciatus* | *Culex quinquefasciatus* | MH423504.1 | 100.00% | 98.15% | Also 98% similarity to *Culex pipiens* |
| Cx_perexiguus_MG_S76 | 621 | *Culex quinquefasciatus* | *Culex perexiguus* | LC473634.1 | 100.00% | 99.52% | Same SNPs to *Culex pipiens* MH374861.1 |
| Ma_uniformis_MG_S77 | 621 | *Mansonia uniformis* | *Mansonia uniformis* | KU187165.1 | 100.00% | 100.00% | |
| Ma_uniformis_MG_S78 | 621 | *Mansonia uniformis* | *Mansonia uniformis* | KU187165.1 | 100.00% | 100.00% | |
| Ma_uniformis_MG_S79 | 626 | *Mansonia uniformis* | *Mansonia uniformis* | KU187157.1 | 100.00% | 99.68% | |
| Cx_poicilipes_MG_S82 | 689 | *Culex bitaeniorhynchus* | *Culex poicilipes* | LC473618.1 | 95.00% | 99.70% | |
| Mi_mediolineata_MG_S83 | 694 | *Culex tritaeniorhynchus* | *Mimomyia mediolineata* | LC473723.1 | 94.00% | 99.39% | |

*Appendix 2—table 1 Continued on next page*

*Appendix 2—table 1 Continued*

| Sequence ID | Sequence length | Morphological identification | BLASTn top hit species | BLASTn top hit accession | Query coverage | % identity | Comments |
|---|---|---|---|---|---|---|---|
| Cx_neavei_MG_S84 | 671 | *Culex tritaeniorhynchus* | *Culex neavei* | LC473635.1 | 98.00% | 99.85% | |
| Ae_scapularis_GF_S85 | 659 | *Aedes scapularis* | *Aedes scapularis* | MN997484.1 | 97.00% | 98.76% | |
| Ae_scapularis_GF_S86 | 658 | *Aedes scapularis* | *Aedes scapularis* | MF172265.1 | 97.00% | 99.38% | |
| Ae_scapularis_GF_S87 | 654 | *Aedes scapularis* | *Aedes scapularis* | MF172265.1 | 98.00% | 99.22% | |
| Ae_serratus_GF_S88 | 660 | *Aedes serratus* | *Aedes serratus* | MF172269.1 | 97.00% | 98.91% | |
| Ae_serratus_GF_S89 | 660 | *Aedes serratus* | *Aedes serratus* | MF172268.1 | 97.00% | 99.22% | |
| Ae_serratus_GF_S90 | 654 | *Aedes serratus* | *Aedes serratus* | MF172268.1 | 98.00% | 99.07% | |
| Cq_venezuelensis_GF_S91 | 658 | *Coquillettidia venezuelensis* | *Coquillettidia venezuelensis* | MN997703.1 | 97.00% | 97.98% | |
| Cq_venezuelensis_GF_S92 | 621 | *Coquillettidia venezuelensis* | *Coquillettidia. venezuelensis* | MN997703.1 | 100.00% | 98.07% | |
| Cq_venezuelensis_GF_S93 | 621 | *Coquillettidia venezuelensis* | *Coquillettidia venezuelensis* | MN997703.1 | 100.00% | 97.75% | |
| Cx_portesi_GF_S94 | 653 | *Culex portesi* | *Culex portesi* | in-house reference library | | 98.5–100% | Reference sequence provided by Amandine Guidez, IP Guyane |
| Cx_portesi_GF_S95 | 693 | *Culex portesi* | *Culex portesi* | in-house reference library | | 98.5–100% | Reference sequence provided by Amandine Guidez, IP Guyane |
| Cx_portesi_GF_S96 | 687 | *Culex portesi* | *Culex portesi* | in-house reference library | | 98.5–100% | Reference sequence provided by Amandine Guidez, IP Guyane |
| Cx_spissipes_GF_S97 | 672 | *Culex spissipes* | *Culex spissipes* | in-house reference library | | 98.5–100% | Reference sequence provided by Amandine Guidez, IP Guyane |
| Cx_spissipes_GF_S98 | 663 | *Culex spissipes* | *Culex spissipes* | in-house reference library | | 98.5–100% | Reference sequence provided by Amandine Guidez, IP Guyane |
| Cx_spissipes_GF_S99 | 660 | *Culex spissipes* | *Culex spissipes* | in-house reference library | | 98.5–100% | Reference sequence provided by Amandine Guidez, IP Guyane |
| Li_durhamii_GF_S100 | 653 | *Lmatus durhamii* | *Limatus durhamii* | MF172330.1 | 98.00% | 99.84% | |
| Li_durhamii_GF_S101 | 621 | *Limatus durhamii* | *Limatus durhamii* | MF172330.1 | 100.00% | 100.00% | |
| Li_durhamii_GF_S102 | 699 | *Limatus durhamii* | *Limatus durhamii* | MF172330.1 | 94.00% | 100.00% | |
| Ma_sp.4_GF_S103 | 621 | *Mansonia titillans* | *Mansonia* sp. | MT329066.1 | 100.00% | 99.84% | 87.12% similarity to *Mansonia titillans* MN968244.1 |
| Ma_sp.4_GF_S104 | 695 | *Mansonia titillans* | *Mansonia* sp. | MT329066.1 | 95.00% | 99.85% | 87.39% to *Mansonia titillans* MN968244.1 |
| Ma_titillans_GF_S105 | 669 | *Mansonia titillans* | *Mansonia titillans* | MN968244.1 | 98.00% | 99.70% | |
| Cx_pedroi_GF_S106 | 653 | *Culex pedroi* | *Culex pedroi* | KX779887.1 | 98.00% | 98.60% | |
| Cx_pedroi_GF_S107 | 661 | *Culex pedroi* | *Culex pedroi* | KX779887.1 | 97.00% | 98.76% | |
| Cx_pedroi_GF_S108 | 621 | *Culex pedroi* | *Culex pedroi* | KX779887.1 | 99.00% | 98.87% | |
| Ps_ferox_GF_S109 | 633 | *Psorophora ferox* | *Psorophora ferox* | MF172349.1 | 100.00% | 99.68% | |

*Appendix 2—table 1 Continued on next page*

*Appendix 2—table 1 Continued*

| Sequence ID | Sequence length | Morphological identification | BLASTn top hit species | BLASTn top hit accession | Query coverage | % identity | Comments |
|---|---|---|---|---|---|---|---|
| Ps_ferox_GF_S110 | 621 | *Psorophora ferox* | *Psorophora ferox* | MF172349.1 | 100.00% | 99.68% | |
| Ps_ferox_GF_S111 | 621 | *Psorophora ferox* | *Psorophora ferox* | MF172347.1 | 99.00% | 99.51% | |
| Ur_geometrica_GF_S112 | 621 | *Uranotaenia geometrica* | *Uranotaenia geometrica* | NC_044662.1 | 100.00% | 100.00% | |
| Ur_geometrica_GF_S113 | – | *Uranotaenia geometrica* | – | – | – | – | No COI obtained |
| Ur_geometrica_GF_S114 | 621 | *Uranotaenia geometrica* | *Uranotaenia geometrica* | NC_044662.1 | 100.00% | 100.00% | |
| Cx_tritaeniorhynchus_MG_S115 | 653 | *Culex tritaeniorhynchus* | *Culex tritaeniorhynchus* | MK861440.1 | 100.00% | 98.77% | |

