## [Editor Report]

Mosquitoes are an important vector for viruses and other pathogens worldwide. However, significant genomic resources are scarce for the study of these species. In this work, the authors create a significant genomic resource that will enable the study of mosquitoes and the pathogens that they carry.

---

## [Decision Letter]

**Decision letter after peer review:**

Thank you for submitting your article " Decoding rRNA sequences for improved metagenomics of sylvatic mosquito species" for consideration by *eLife*. Your article been reviewed by three peer reviewers, one of whom is a member of our Board of Reviewing Editors, and the evaluation has been overseen by a Senior Editor. The following individuals involved in review of your submission have agreed to reveal their identity: Leslie B Vosshall (Reviewer #1); Katherine I Young (Reviewer #2)

Essential Revisions:

The paper is very nice, I just need you to make some improvements in the presentation, particularly to the figures and legends. Please address all of the comments of Reviewers 1 and 3. Please work hard to make your figure fonts as legible as possible, and to provide color keys and annotations on the phylogenies that make them as accessible as possible.

The paper also needs a better evaluation of which is the superior phylogenetic approach. I agree with reviewer 3 that a table or figure comparing the two phylogenies (rRNA vs COI) would be very helpful.

It feels like you are saying that the phylogeny created by rRNA is better than the one by COI? If this is true, please directly state this in the abstract and elsewhere.

Congratulations on your excellent work.

*Reviewer #3 (Recommendations for the authors):*

– The main confusion that I have after reading this paper is – how do the rRNA and COI trees compare? One has to study the trees, and long paragraphs of discussion of them, to attempt to understand where they are the same and where they are different. (to be honest, the small fonts were exhausting and I gave up trying) Could some sort of table or graphic summarize this better? Maybe a table with columns for COI-placement and rRNA- placement, with rows for each species, and colors in the table showing where the placements are the same? This might help the reader understand where the methods produce the same results and where they are different.

– The paper is very well-written, but in contrast the figures and figure legends need more attention. Some figures do not have titles; figure legends do not have enough information to interpret figures; fonts are too small on figures; and color-keys should be given on the figures themselves as well as written in the legend. Importantly, why are there Genbank sequences on some of the trees and not others? Even where there are, I can't see the difference between bold and not-bold.

– Please carefully double-check Table 1. In particular, please confirm the information in the "vector for" column, and confirm that the cited references actually support those host-virus relationships. As an example, I could not find clear evidence of Cx quinquefasciatus being a vector for DENV in any of the 3 references cited for that row.

---

## [Author Response]

Essential Revisions:The paper is very nice, I just need you to make some improvements in the presentation, particularly to the figures and legends. Please address all of the comments of Reviewers 1 and 3. Please work hard to make your figure fonts as legible as possible, and to provide color keys and annotations on the phylogenies that make them as accessible as possible.

All comments from Reviewer 3 have been addressed.

Colour keys with full genus names were added and font sizes were increased in all phylogeny figures.

The paper also needs a better evaluation of which is the superior phylogenetic approach. I agree with reviewer 3 that a table or figure comparing the two phylogenies (rRNA vs COI) would be very helpful.

We have added two tables to address this. Table 2 in the Results section presents the differences in the rRNA and COI phylogenies. Table 3 at the end of the Discussion section evaluates the advantages and disadvantages of each approach.

It feels like you are saying that the phylogeny created by rRNA is better than the one by COI? If this is true, please directly state this in the abstract and elsewhere.

We have added statements affirming that rRNA phylogeny is better than COI phylogeny, on the basis that it reflects current mosquito systematics more closely, in the Abstract (line 33–36), at the end of the Introduction (lines 107–109) and in the Discussion (lines 647–655).

Reviewer #3 (Recommendations for the authors):– The main confusion that I have after reading this paper is – how do the rRNA and COI trees compare? One has to study the trees, and long paragraphs of discussion of them, to attempt to understand where they are the same and where they are different. (to be honest, the small fonts were exhausting and I gave up trying) Could some sort of table or graphic summarize this better? Maybe a table with columns for COI-placement and rRNA- placement, with rows for each species, and colors in the table showing where the placements are the same? This might help the reader understand where the methods produce the same results and where they are different.

Table 2 has been added to the Results section summarising how the rRNA and COI phylogenies differ from each other.

– The paper is very well-written, but in contrast the figures and figure legends need more attention. Some figures do not have titles; figure legends do not have enough information to interpret figures; fonts are too small on figures; and color-keys should be given on the figures themselves as well as written in the legend. Importantly, why are there Genbank sequences on some of the trees and not others? Even where there are, I can't see the difference between bold and not-bold.

Figure titles have been added to all figures. Figure legends have been expanded with additional detail. Font sizes on phylogeny figures have been increased and colour keys are now provided.

Figure 3 features a tree of full-length 28S rRNA sequences originating from this study and from existing GenBank records. Sequences from GenBank from other published studies have accession numbers while those from this study do not.

Sequences from GenBank were included as a “positive control” in the alignment analysis together with our 28S rRNA sequences to demonstrate that our sequences cluster correctly by taxonomy alongside published sequences. In Figure 5, we used COI sequences of three water mite species obtained from GenBank to act as an outgroup as we were unable to obtain *Horreolanus* COI sequences from our own specimens.

GenBank sequences are now annotated with filled circles in Figures 3 and 5.

– Please carefully double-check Table 1. In particular, please confirm the information in the "vector for" column, and confirm that the cited references actually support those host-virus relationships. As an example, I could not find clear evidence of Cx quinquefasciatus being a vector for DENV in any of the 3 references cited for that row.

Additional references have been added for the *Culex quinquefasciatus* row. Table 1 has been double-checked.